# New Data for the Internal Structure of Ultramafic Hosted Seafloor Massive Sulfides (SMS) Deposits: Case Study of the Semenov-5 Hydrothermal Field (13°31′ N, MAR)

**Anna Firstova** [1,*] , **Georgy Cherkashov** [1,2,*], **Tamara Stepanova** [1], **Anna Sukhanova** [1], **Irina Poroshina** [1] **and Victor Bel'tenev** [1]

[1] Institute for Geology and Mineral Resources of the Ocean (FSBI "VNIIOkeangeologia"), 1 Angliisky Ave., 190121 St. Petersburg, Russia

[2] Institute of Earth Sciences, St. Petersburg State University, 7/9 Universitetskaya Emb., 199034 St. Petersburg, Russia

[*] Correspondence: anetfirst@gmail.com (A.F.); gcherkashov@gmail.com (G.C.); Tel.: +8-981-828-99-73 (A.F.); +7-812-713-8378 (G.C.)

**Abstract:** The internal structure of Seafloor Massive Sulfides (SMS) deposits is one of the most important and complex issues facing the study of modern hydrothermal ore systems. The Semenov-5 hydrothermal field is a unique area where mass wasting on the slope of the oceanic core complex (OCC) structure exposes the subsurface portion of the deposit and offers an exceptional opportunity to observe massive sulfides that have formed not only on the seafloor but in sub-seafloor zones as well. This paper examines the internal structure of the OCC-related Semenov-5 hydrothermal field along with analysis of the mineralogy and chemistry of different parts of sulfide deposit. The seafloor deposit is comprised of pyrite, marcasite, hematite, goethite, lepidocrocite, rare pyrrhotite, isocubanite and Co-rich pyrite. Sulfide chemistry indicates the prevailing influence of ultramafics on their composition irrespective of the spatial relation with basalt lavas. Sub-seafloor mineralization is associated with ultramafic rocks and is represented by massive and disseminated sulfides. Pyrrhotite, isocubanite, pyrite, chalcopyrite, Co-rich pyrite, quartz with rutile, quarts with hematite and Cr-spinels are fixed in massive subseafloor mineralization. The presence of Cr-spinels as well as a very high Cr content are regarded as indicators of the metasomatic nature of this part of the deposit that had formed as a result of ultramafic replacement. As a result, three zones of a hydrothermal ore-forming system have been described: massive sulfides precipitated from hot vents on the surface of the seafloor, massive sulfides formed due to replacement of ultramafics below the seafloor and disseminated sulfide mineralization-filled cracks in hosted rocks which have formed stockwork around metasomatic massive sulfides. Despite differences in the mineral and geochemical composition of sub-seafloor and seafloor mineralization, all minerals subject to the sample formed as a consequence of fluid circulation in ultramafic rocks and were linked by a common ore-forming process.

**Keywords:** seafloor massive sulfides; Mid-Atlantic Ridge; hydrothermal processes; mass wasting landslide processes; reconstruction model

## 1. Introduction

One of the most complicated issues in the study of seafloor massive sulfides (SMS) is the modeling of the internal structure of ore bodies and the sequencing of their formation. The structure of the hydrothermal ore system is determined by the geological setting that involves the structural and tectonic characteristics of the area and the composition of host rocks. Within the Northern Equatorial Mid-Atlantic Ridge (NEq MAR), up to half of the hydrothermal fields and SMS deposits are related to tectonic segments with deep-seated gabbro-peridotite rocks exposed at the slopes of rift valley. The ultramafic hosted fields (from North to South), Logatchev, Semenov, Irinovskoye, Ashadze and the recently

discovered Molodezhnoe and Korallovoe, are distributed along the NEq MAR from 13° to 15° N (Figure 1A).

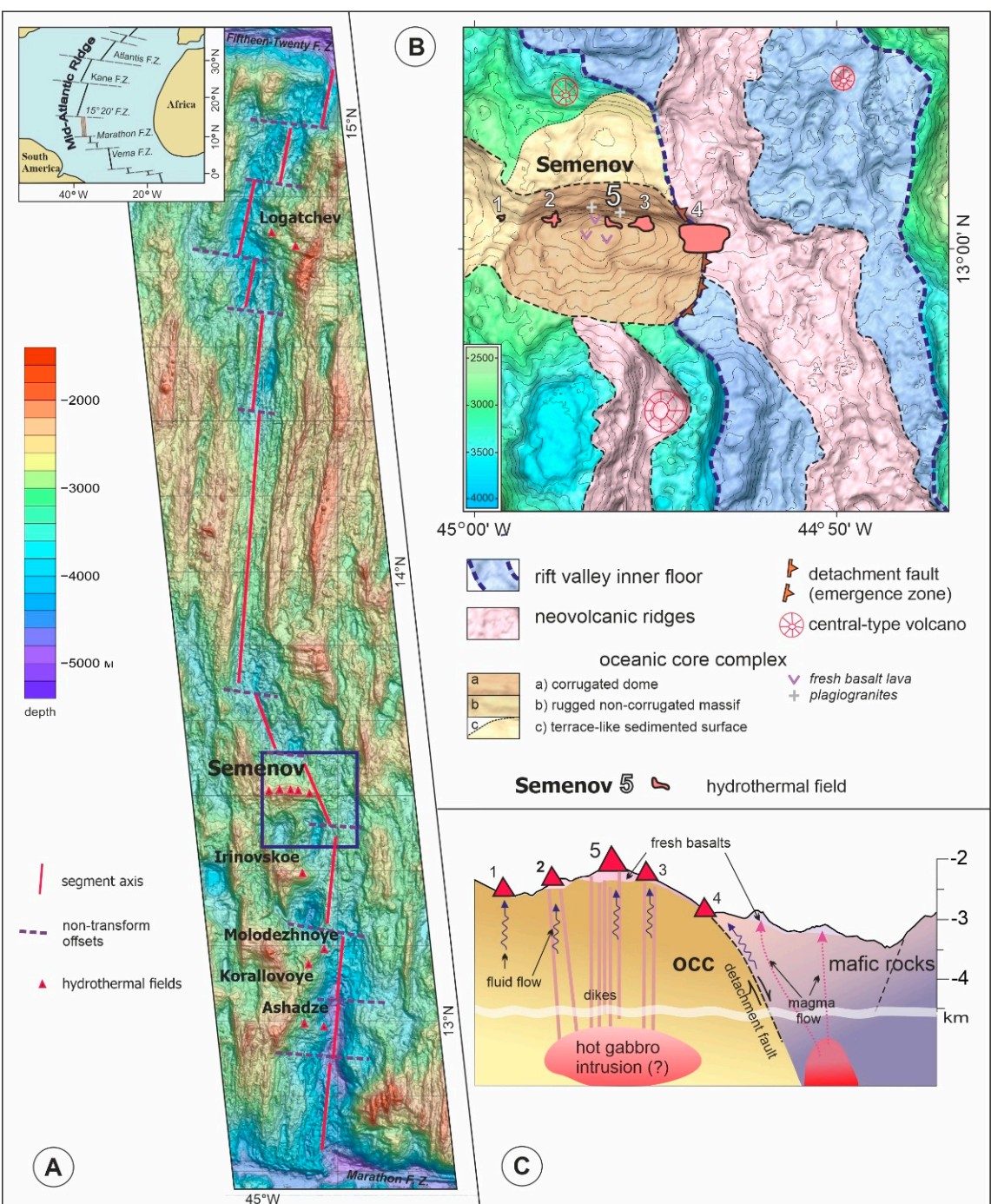

**Figure 1.** Semenov hydrothermal area. (**A**) The regional bathymetry of the MAR segment between 15–20 and Marathon fracture zones and the location of the hydrothermal fields. The box corresponds to the area of (**B**). (**B**) Morphostructural setting of the Semenov hydrothermal fields. (**C**) Interpretative cross-section of the MAR rift valley in the Semenov cluster (modified from Pertsev et al. [1]); vertical exaggeration 3:1.

Up until now, knowledge of the internal structure of SMS deposits is based mainly on drilling results and is very scarce compared to investigations of the internal structure of land deposits [2]. There are data from the International Ocean Discovery Program (IODP),

BLUE MINING and Nautilus Minerals which present the 3rd dimension of SMS deposits related to basalt hosted within MAR and arc-back-arc settings [3–9]. However, there is no direct information about internal structure for ultramafic-hosted deposits Geophysical data that can provide indirect information about this issue are also rare for them [10]. Direct observation of the internal structure of ore bodies is virtually nonexistent. Mass wasting landslides can offer an exceptional opportunity to observe the surface part of an ore body that is composed of massive sulfides and the subsurface zone as well. Such an opportunity arose in the area of the Semenov-5 (S-5) hydrothermal field at 13°31′ N, where a slope landslide led to the exposure of the subsurface part of the deposit. The mineralogy and chemistry of sulfides recovered from the outcropped zone allowed for modeling its inner structure and for reconstructing the ore-forming processes involved in an ultramafic-hosted SMS deposit. Here, we present the first detailed data related to the mineralogy and geochemistry of massive sulfides for Semenov-5.

## 2. Previous Study and Geological Setting of the Semenov-5 Hydrothermal Field

The Semenov-5 hydrothermal field, as a part of the Semenov cluster, was discovered in 2009 during the cruise of the R/V Professor Logatchev [11], two years after the discovery of the first four Semenov fields in 2007 [12–14]. The study of the Semenov deposits between 2007 and 2009 included sampling by TV-grab and dredge, video-profiling and SP measurements. Bathymetry was provided by hull-mounted multibeam echosounding. Several papers related to mineralogy, chemistry and sulfide dating resulted from these expeditions [14–19].

The next study of the Semenov area, in 2013, was conducted during the ODEMAR cruise onboard the N/O Pourquoi Pas (IFREMER, France) [20]. Microbathymetry data for the area was collected by an autonomous vehicle (AUV) and geologic observations and sampling from a towed vehicle (ROV). Based on this data, the tectonic structure, evolution and the nature of oceanic core complexes were presented in a paper by J. Escartin et al. [20]. Landslide processes at the northern slope of the dome structure were also detected during this cruise.

The Semenov deposits was recently visited in 2022 on the R/V James Cook (SOC, UK) as part of the ULTRA program [21]. In this expedition, visual observations from AUV and ROV, geophysical surveys, and drilling were carried out. The results of this cruise have not yet been published.

The Semenov hydrothermal area is located at a latitude of 13°30′ N on the western flank of the MAR second order segment between the Fifteen-Twenty and Marathon fracture zones (Figure 1A) and represents a smoothed crest of an east–west elongated corrugated dome-shaped structure identified as an oceanic core complex (OCC) (Figures 1B,C and 2) [1,14,22–25].

The OCC formation is attributed to long-term slips with exhumation of deep-seated rocks along large-scale detachments [22–29]. The 13°30′ N OCC detachment surface is disrupted by normal faulting, fissuring and mass wasting (Figure 2). The corrugated dome is composed mainly of peridotites/serpentinites, with a minor occurrence of gabbro. It is partially covered by a relatively thin flows of basalts, e.g., samples of very fresh basalt lavas as well as plagiogranites that were recovered from the summit surface [1,14,20].

The Semenov-5 field lies on the northern slope of the dome structure at depths of 2200–2250 m (Figure 2b) and is hosted by hydrothermally altered (serpentinized) gabbro-peridotites and fresh basalts. The dimensions of the field are estimated by video profiling to be 1200 × 650 m. No signs of current hydrothermal activity are detected. The dome surface at the southern part of the S-5 area is affected by mass wasting, with coalescing crescent-shaped and steep slump scarps associated with debris deposits downslope [20].

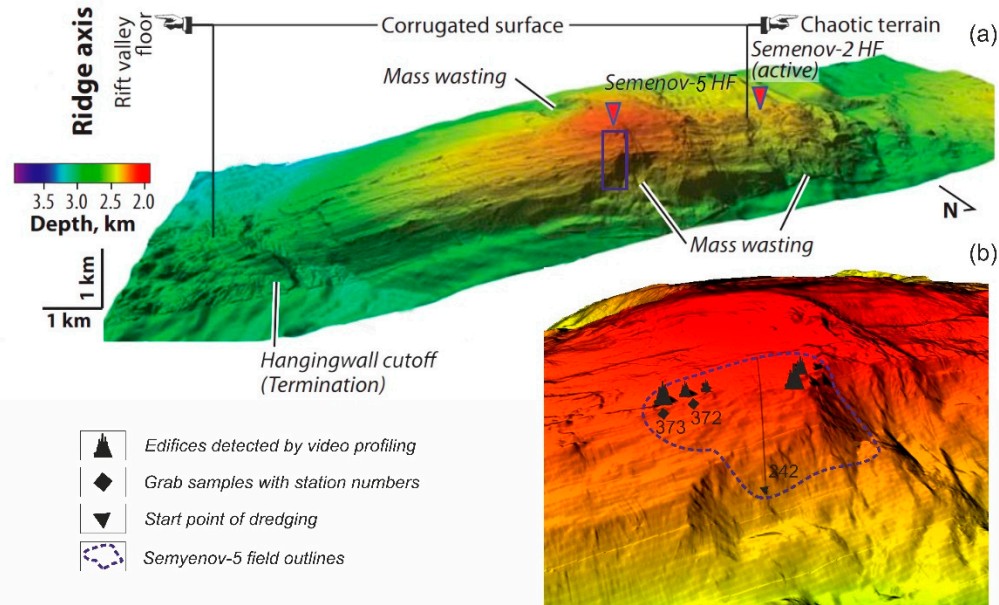

**Figure 2.** The 13°30′ N OCC detachment surface. (**a**) Three-dimensional views of the 13°30′ N OCC microbathymetry showing the different structural domains together with other morphologic features according to ODEMAR data [20]. The box corresponds to the area of (**b**). (**b**) Location of the Semenov 5 field at the northern slope of OCC and position of TV-Grab and dredge stations. Field outlines are determined by video profiling [12,13].

## 3. Materials and Methods

In total, 55 kg of massive and disseminated sulfides were recovered by TV grab (st. 372, 373) and by dredge (st. 242) and initially described on board RV Professor Logatchev in 2007.

The major and minor mineralogical phases were identified using reflected light microscopy. Whole rock chemistry analysis for 20 subdivided samples was performed at VNIIOkeangeologia, St. Petersburg, Russia.

$SiO_2$ was analyzed by a standard photometric method using a Shimadzu uv-1650-pc spectrometer (Shimadzu, Kyoto, Japan). Samples were decomposed by melting with sodium carbonate. $SiO_2$ content was estimated using the product of reduction of the yellow complex of silicon-molybdenum heteropoly-acid. Ascorbic acid is a reducing agent. Concentration of S is determined by gravimetric method and precipitation with $BaCl_2$.

Elements such as Fe, Cu, Zn, Pb, Cd, Ag, Ca, Mg and Al were determined by atomic absorption spectroscopy (brand C-155 spectrometer with flame atomization). Quality control at the laboratory of VNIIOkeangeologia was performed by analysis of state standard (GSO) of water metal solutions (7254-96-Fe, 7255-96 Cu, 7256-96 Zn, 7252-96 Pb, 7472-98 Cd, 842-2002 Ag).

Minor geochemical elements Co, Ni, Bi, Se, Te, In, Ge, Ga, Sb, As, Sn, Mo, V, Cr, U and Ba were analyzed with an Agilent 7700 quadrupole inductively coupled plasma-mass spectrometer at VSEGEI, St. Petersburg, Russia. The test portion of a sample was 100 mg. Samples were digested in high-density graphite or glass carbon autoclaves using 5 mL $HNO_3$ + 5 mL $HClO_4$ and 10 mL HF at 180 °C. All the pure acids used for digestion were additionally purified in a BSB-939-IR apparatus (Berghof, Germany). The water for dilution was deionized in a DEMIWA 20-100 roa Watek Water Purification System (Watek, Ledeč nad Sázavou, Czech Republic). Both instruments were calibrated with references provided by the spectrometer manufacturers. Quality control was based on analysis of control samples from the International Geoanalytical Proficiency Testing Program, complying with ISO 17025 standards. To quality check the precision of the method, a state-certified sample of RUS-4 and a sample of SdAR-L2 (USA) from round 37A of the International Geoanalytical

proficiency-testing program were used. The enlarged uncertainties (K = 2) of precision for all analyses did not exceed 30%.

The whole rock Au content in 15 samples was determined by Atomic Absorption Spectroscopy (AAS) at the Central Laboratory of VSEGEI with the aid of a Perkin Elmer Analyst-800 spectrometer. Quality control was performed by analysis of the state standard (GSO) of flotation concentrate for Au-bearing sulfides CZK-3 (2739-83 Au).

Duplicate analyses were performed on 10% of the samples and the average error did not exceed ±1% for all techniques.

Major element analyses of ore-forming minerals were carried out on a Hitachi S3400N scanning electron microscope (SEM) equipped with an AzTec Energy 350 energy dispersive X-ray spectroscopy (EDX) detector and an INCA 500 wavelength dispersive X-ray spectroscopy (WDX) detector based at the Geomodel Center (St. Petersburg State University, Saint Petersburg, Russia) using an acceleration potential of 20 kV, a beam current of 2 or 10 nA and a spot size from 3–5 μm for the EDX and WDX procedures, respectively. The following standards were used: Bi, Te, Au, Ag, Ni, Cu, In, As, Sb, Zn, Fe, FeS$_2$ and Mo standards by Geller microanalytical laboratory and Bi, Se, FeS$_2$ and CaSO$_4$ by MAC (Micro-Analysis Consultants, Ltd., Cambridgeshire, UK). The thin-sections were carbon-coated for SEM analysis and imaged using backscattered and secondary electrons (BSE).

## 4. Results

Geological samples were recovered from different parts of the Semenov-5 deposit. The dredge at st. 242 crossed the outcropped landslide zone in the northern part of the field from the bottom of the slope towards to top and collected massive sulfides, vein-disseminated mineralization and altered gabbro-peridotites (gabbroids, serpentinites). Massive sulfides with basalts were sampled from the southern upper zone close the top of the dome (st. 372 and 373) (Figure 2b). Results of mineralogical and chemical analysis of samples from all stations are presented below.

### 4.1. Mineralogy

Samples recovered from the Semenov-5 hydrothermal field are characterized by a large variety of sulfides and other minerals (Table 1).

**Table 1.** Mineral composition of samples from Semenov-5 hydrothermal field.

| Samples | 242-1 | | | | | | 242-2 | | | | 372 | 373 | | | |
|---|---|---|---|---|---|---|---|---|---|---|---|---|---|---|---|
| Zones | Py Layer | Contact of Sulfides with the Rock | | | Py-Chp Layer | | Contact of Sulfides with the Rock | Py-Chp Layer | | | | | | | |
| Minerals | 1 | 1/1 | 1/1a | 1/1b | 1/2 | 1/3 | 2/3 | 2 | 2/1 | 2/2 | 1 | 1 | 3 | 3/1 | 4 |
| Pyrrhotite Fe$_{1-x}$S | | | | | | | | x | | | | | | | |
| Pyrite FeS$_2$ | +++ | +++ | +++ | +++ | +++ | +++ | +++ | +++ | +++ | +++ | +++ | +++ | +++ | +++ | +++ |
| Marcasite FeS$_2$ | | x | | x | x | | | | | | + | +++ | ++ | +++ | +++ |
| Chalcopyrite CuFeS$_2$ | + | + | ++ | ++ | +++ | +++ | ++ | +++ | +++ | +++ | | x | x | x | x |
| Isocubanite CuFe$_2$S$_3$ | | | | | | | | | | | | | | ? | x |
| Secondary Cu sulfides (Bn,Cv) | | | | | | | x | x | | | | x | x | x | x |
| Sphalerite (Zn,Fe)S | x | | x | | | | | | | | | | x | + | |
| Cr-spinels (Fe,Mg)(Cr,Al,Fe)$_2$O$_4$ | x | x | x | x | x | x | x | x | x | x | | | | | |
| Rutile TiO$_2$ | | | | | | x | x | x | x | x | | | | | |
| Quartz SiO$_2$ | + | + | + | + | + | + | +++ | ++ | ++ | ++ | | | | | |
| Baryte BaSO$_4$ | | | | | | | | | | | x | + | + | + | ++ |
| Hematite Fe$_2$O$_3$ | | | | | | | | | | | x | ++ | | + | ++ |
| Goethite FeO(OH) | | | | | | | | | | | x | + | + | + | ++ |
| Lepidocrocite γ-Fe$^{3+}$O(OH) | | | | | | | | | | | | | + | | |

Notes: +++ major minerals, ++/+ minor minerals, x rare minerals.

The description of sulfide mineralogy successively for three stations (242, 372 and 373) is presented below.

### 4.1.1. St. 242

Massive sulfides from st. 242 were represented by two samples. The main feature of these samples is the well-defined contact of massive sulfide with altered host-rock (Figure 3). The major sulfide minerals are pyrite and chalcopyrite; minor minerals include pyrrhotite, marcasite and sphalerite.

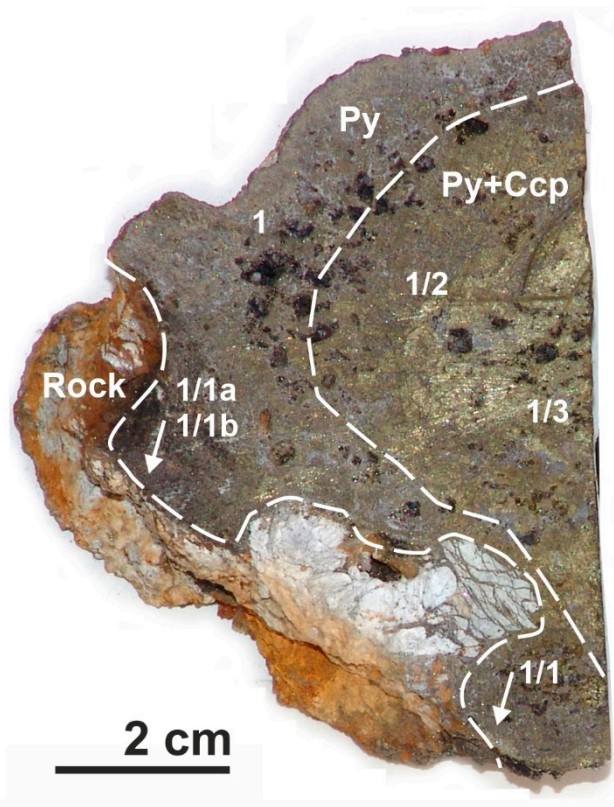

**Figure 3.** A general view of sample 242-1 with zonal texture.

Sample 242-1 has a distinct zonal texture with altered rock and massive sulfides (Table 1, Figure 3). The hydrothermally altered rock is partially or completely leached. The massive sulfide is represented by pyrite and chalcopyrite-pyrite layers. At the contact of sulfides with rock, a porous-layered aggregate of thin quartz-pyrite veinlets is identified.

The pyrite layer (up to 7 cm) is comprised of porous, dendritic pyrite. Pyrite is fine-grained in the center of the dendrites and course-grained at the edges. A number of structures of pyrrhotite tabular replacement appear only as contours in pyrite or incompletely filled tabular formations (Figure 4a). "Bird's eye" textures were found in this layer (Figure 4b).

Closer to the next layer, pyrite aggregates are composed of intergrown subhedral zonal crystals (Figure 4c). On the walls of voids, crystals are usually euhedral. The smallest grains are sphalerite (from 0–0.5 mm) and sometimes traced along zones in pyrite. Rare small grains of chalcopyrite were also detected. Fragmented grains of Cr-spinels are constantly presented in pyrite. Cr-spinels exhibit traces of alteration: border rims are observed along the cracks, differing in color and composition (Figure 4d). The changing in Cr-spinels' composition will be described below. A disseminated xenomorphic quartz does exceed 2% in this zone. The composition of sulfide minerals are given in Table 2.

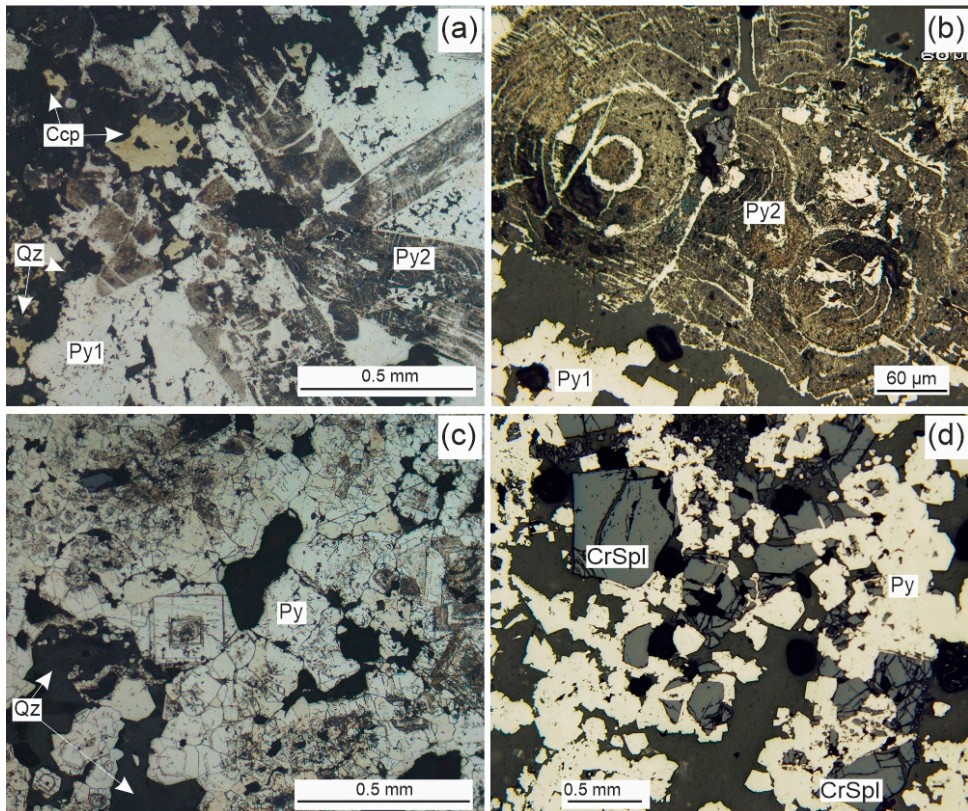

**Figure 4.** Microstructures of sample 242-1. (**a**) Pyrite (Py1), trace of tabular pyrrhotite completely replaced by pyrite, (Py2) and chalcopyrite (sample 242-1). (**b**) "Bird's eye" structure in composed by Py2 (sample 242-1/1). (**c**) Zoned pyrite crystals (sample etched with concentrated nitric acid with fluorite powder) (sample 242-1/1b). (**d**) Fragmented grains of Cr-spinels in pyrite (sample 242-1/1).

**Table 2.** Average chemical composition of sulfides from st. 242.

| Mineral (Number of Analyses, *n*) | Element (wt %) | | | | | | |
|---|---|---|---|---|---|---|---|
| | **Fe** | **Cu** | **Zn** | **S** | **O** | **Si** | **Total** |
| Fe-sphalerite (39) | 10.27 | - | 56.12 | 33.40 | - | - | 99.79 |
| Chalcopyrite (47) | 31.37 | 33.68 | - | 34.49 | - | - | 99.54 |
| Secondary Cu sulfides (18) | 2.32 | 57.13 | - | 32.54 | 8.20 | - | 100.19 |
| Quartz (27) | - | - | - | - | 50.93 | 48.92 | 99.85 |

Notes: - not detected.

The chalcopyrite-pyrite layer (up to 10 cm) is comprised of xenomorphic fine-medium-grained chalcopyrite intergrown with round, lumpy, fine-grained pyrite aggregates (Figure 5a). Coarse-grained (up to 0.5 mm) and subhedral chalcopyrite forms rims of channels where the smallest impregnation of primary unaltered pyrrhotite can be occasionally observed (Figure 5b).

There are relics of pyrrhotite in pyrite accumulations up to 5 mm in size. The pyrite is relatively coarse-grained, euhedral and cubic around a canal. Quartz (up to 5 vol. %) is scattered, fine-grained and often euhedral. Large, fragmented Cr-spinel crystals occur sporadically. Cr-spinels alter at different degrees. Alteration and border rims develop partly in the central area. The most strongly altered fragments of Cr-spinels crystals have a corroded, skeletal surface (Figure 6a).

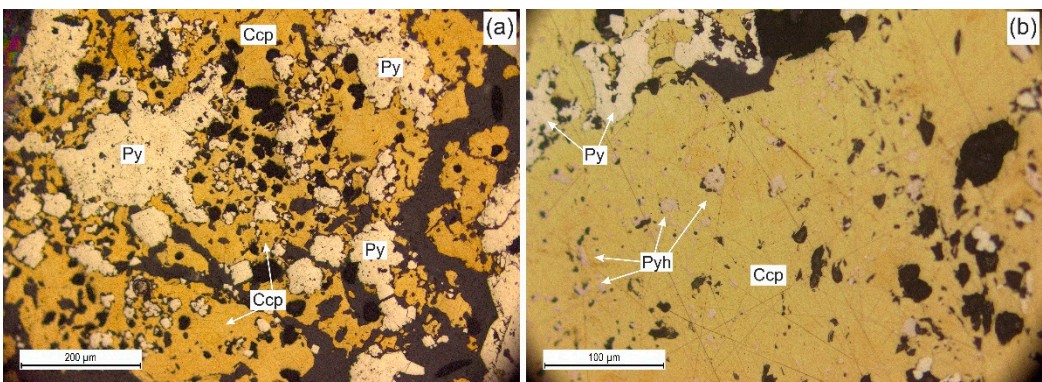

**Figure 5.** Chalcopyrite-pyrite layer in sample 242-1. (**a**) Chalcopyrite-pyrite intergrowths (sample 242-1/2). (**b**) Massive aggregate of coarse-grained chalcopyrite with primary unaltered pyrrhotite (sample 242-1/2).

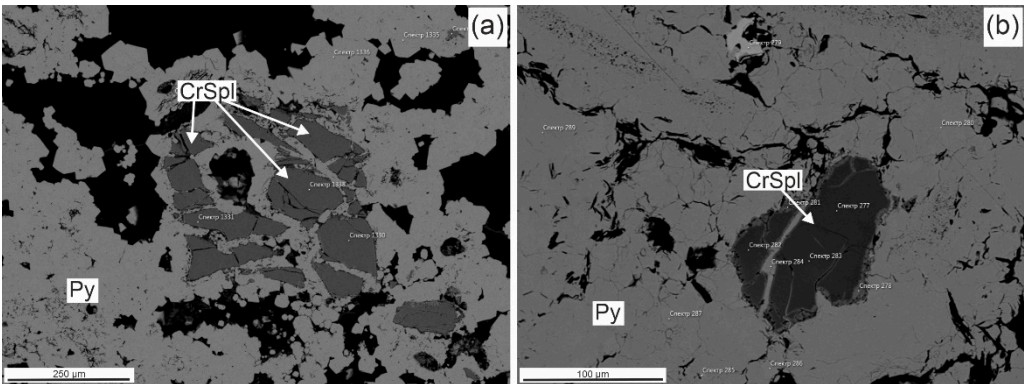

**Figure 6.** Cr-spinels in sulfide fragment 242-1. BSE images. (**a**) Cr-spinel intergrowths with Ni-Co pyrite. (**b**) Cr-spinels with alteration rims.

Pyrites from pyrite and chalcopyrite-pyrite layers have a different chemical composition (Table 3).

**Table 3.** Variations in the pyrite compositions from st. 242 (wt %).

| | Element (wt %) | | | | | | | |
|---|---|---|---|---|---|---|---|---|
| | **S** | **Fe** | **Co** | **Ni** | **Cu** | **Zn** | **Cr** | **Total** |
| | Pyrites from pyrite layer, *n* = 198 | | | | | | | |
| Average | 52.78 | 47.02 | 0.43 | 0.47 | 0.35 | - | 0.32 | 100.05 |
| Min | 45.92 | 45.40 | 0.01 | 0.05 | 0.04 | - | 0.27 | 98.69 |
| Max | 54.97 | 54.08 | 1.08 | 1.11 | 1.16 | - | 0.37 | 101.77 |
| Frequency of occurrence in a layer | 100 vol.% | 100 vol.% | 15 vol.% | 14 vol.% | 13 vol.% | - | <1 vol.% | |
| | Pyrites from chalcopyrite-pyrite layer, *n* = 297 | | | | | | | |
| Average | 52.27 | 46.17 | 0.78 | 0.56 | 0.71 | 0.37 | 1.45 | 100.31 |
| Min | 47.37 | 42.37 | 0.02 | 0.04 | 0.08 | 0.18 | 0.32 | 98.38 |
| Max | 54.04 | 50.36 | 2.97 | 2.01 | 4.80 | 0.56 | 4.06 | 101.80 |
| Frequency of occurrence in a layer | 100 vol.% | 100 vol.% | 62 vol.% | 61 vol.% | 63 vol.% | <1 vol.% | <1 vol.% | |

Notes: - not detected.

The pyrites of the pyrite layer are mostly characterized by close to stoichiometric composition. Only 13%–15% of all pyrite analyses from the pyrite layer contain Co (0.01–1.08 wt %), Ni (0.05–1.11 wt %) and Cu (0.04–1.16 wt %). In the chalcopyrite-pyrite layer, impurities in pyrites are more common (61%–63% of all pyrite analyses) and the

contents of Co, Ni and Cu are much higher: 0.02–2.97, 0.04–2.01 and 0.08–4.80 wt %, respectively. Another impurity in pyrites of both layers is Si, whereas Zn and Ti were found in the chalcopyrite-pyrite layer as well.

The sample 242-2 has a structure similar to 242-1 with some differences (Figure 7). The pyrite layer is much thinner and there are no pyrite tabular crystals replacing pyrrhotite with a "birds-eye" structure. The layer is represented by accumulations of porous-layered thin quartz-pyrite veinlets. The transition from pyrite to chalcopyrite-pyrite layer is gradual.

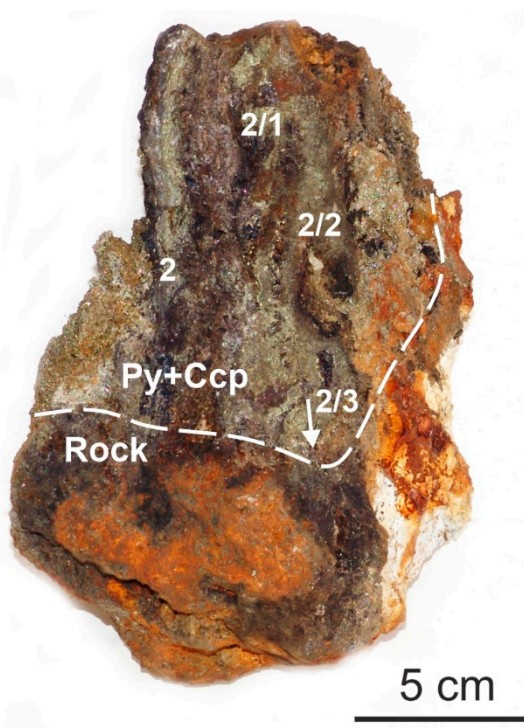

**Figure 7.** A general view of sample 242-2 composed of pyrite, chalcopyrite and altered rock.

The chalcopyrite-pyrite layer has a structure quite similar to the 242-1 sample described above. Moreover, there is also isocubanite with a chalcopyrite exsolution (Figure 8a).

On rare occasions, isocubanite is replaced by porous pyrite. (Figure 8b). Chalcopyrite around a void (assuming as microchannel for supplying hydrothermal fluid) has an exsolution structure and primary pyrrhotite grains (Figure 8c). The main difference is the abundance of quartz (up to 25 vol. %). Dipyramidal euhedral quartz crystals are disseminated in the sample and accumulate, forming rims around the o-shaped voids (assuming as microchannel for hydrothermal fluid). There are a lot of acicular rutile (Figure 8d) and spherulite of hematite (Figure 8e). Euhedral cubic pyrite is related to quartz accumulations. Acute-angled fragments of Cr-spinels are present everywhere (Figure 9a,b).

The vein-disseminated mineralization in sample 242 is represented by thin quartz-pyrite and pyrite veins (Figure 10), as well as dissemination of pyrite, pyrite-sphalerite intergrowths and Cr-spinels in hydrothermally altered ultramafic rock (Figure 11). Cr-spinels have a border rim differing in color and composition. The chemical composition of minerals is represented in Table 4. Based on microprobe analysis, the altered ultramafic rocks are made up of enstatite (MgO = 31.5%, $SiO_2$ = 65.3%, $Fe_2O_3$ = 3.2%).

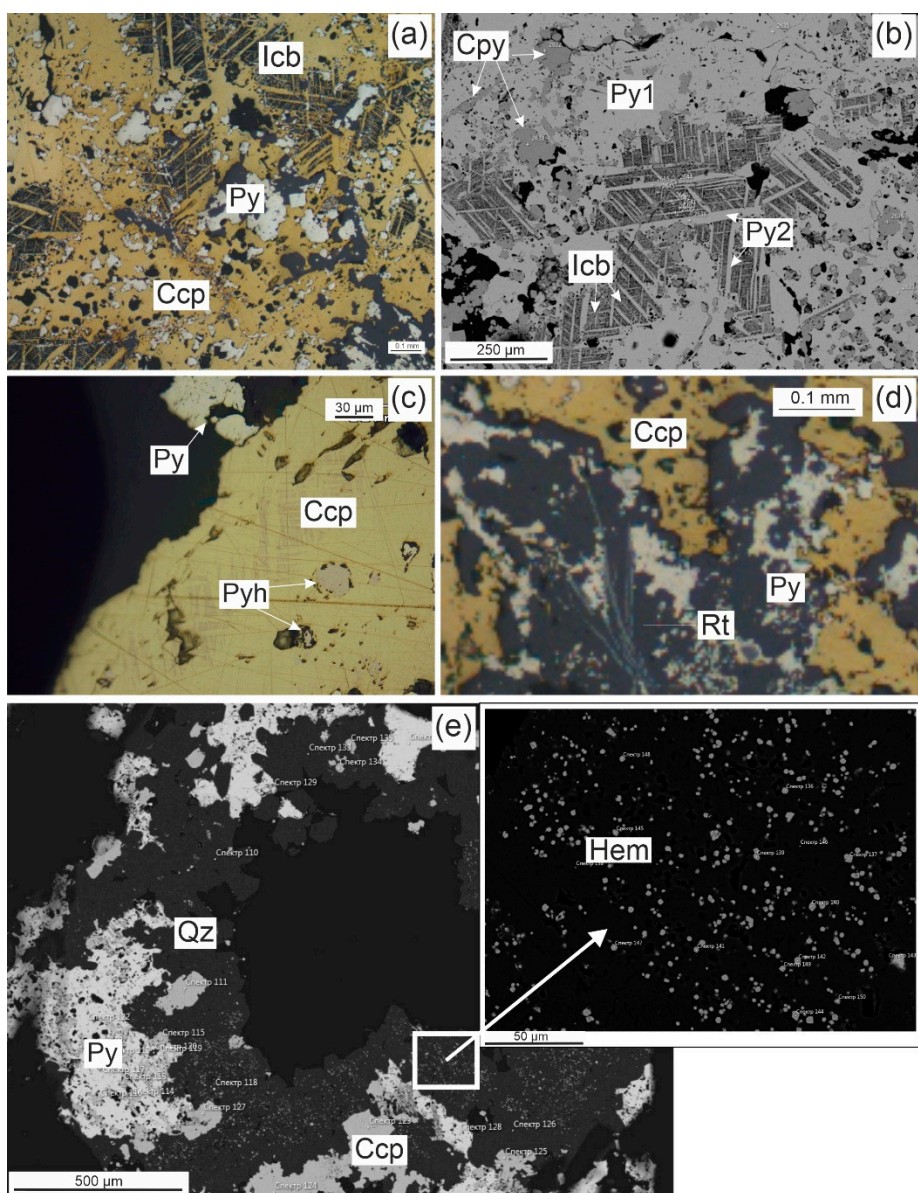

**Figure 8.** Chalcopyrite-pyrite layer in sample 242-2. (**a**) Lattice isocubanite in chalcopyrite grains (sample 242-2/2). (**b**) Porous pyrite (Py1) and lattice pyrite (Py2) formed after leached isocubanite. BSE image. (**c**) Lattice texture of exsolved isocubanite with chalcopyrite. Primary pyrrhotite in chalcopyrite around a void (sample 242-2). (**d**) Rutile in quartz (sample 242-2/1). (**e**) Hematite grains in quartz around a void. BSE image.

**Table 4.** Average chemical composition of minerals from vein-disseminated samples.

| Mineral (Number of Analyses, *n*) | Element (wt %) | | | | | | | | | | | |
|---|---|---|---|---|---|---|---|---|---|---|---|---|
| | Fe | Cu | S | Ni | Co | O | Si | Mg | Al | Cr | Mn | Total |
| Pyrite (11) | 45.66 | - | 52.72 | 1.41 | 1.29 | - | - | - | - | - | - | 101.08 |
| Chalcopyrite (5) | 30.91 | 32.98 | 34.01 | 0.43 | 0.24 | 2.92 | - | - | - | - | - | 101.49 |
| | Oxides (wt %) | | | | | | | | | |
| | FeO | NiO | CoO | SiO$_2$ | MgO | Al$_2$O$_2$ | Cr$_2$O$_3$ | MnO | Total |
| Enstatite (5) | 3.33 | - | - | 66.45 | 31.25 | - | 0.22 | - | 101.25 |
| Cr-spinels. Core (3) | 15.06 | - | - | - | 13.48 | 28.11 | 38.57 | 6.48 | 101.70 |
| Cr-spinels. Rims (2) | 63.81 | 0.87 | 0.70 | 0.74 | 0.90 | 2.15 | 25.73 | 6.42 | 101.32 |

Notes: - not detected.

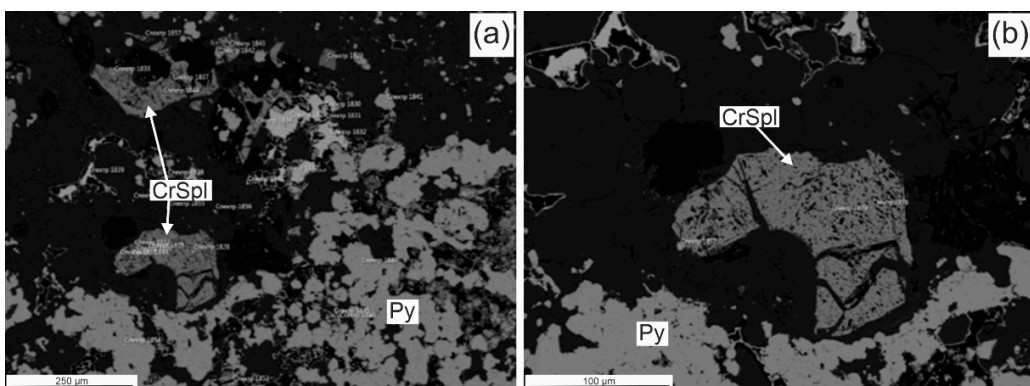

**Figure 9.** Porous Cr-spinels in sulfide sample 242-2. BSE images. (**a**) Acute-angled fragments of Cr-spinels in porous pyrite. (**b**) Acute-angled grain of Cr-spinels.

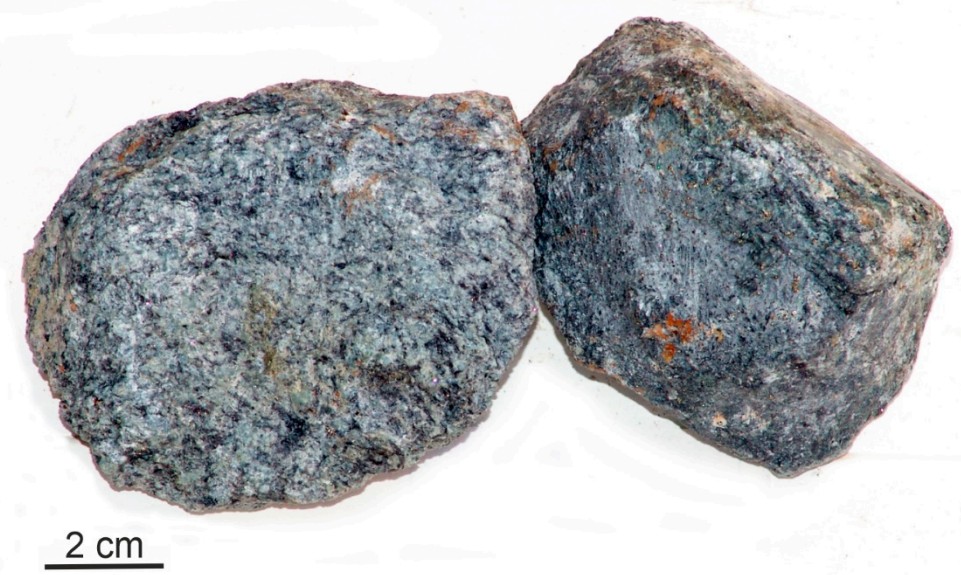

**Figure 10.** A general view of sample 242 with vein-disseminated mineralization.

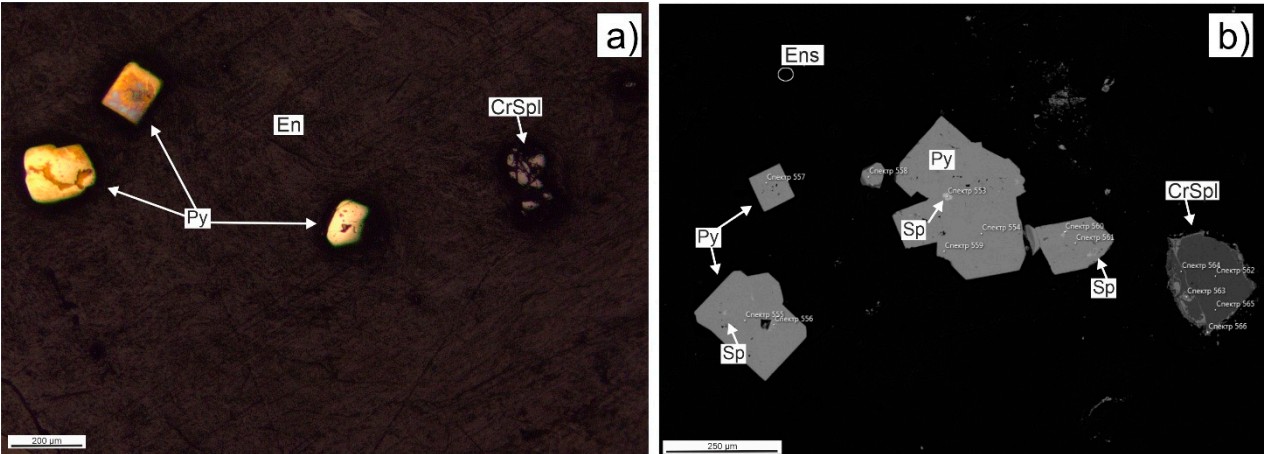

**Figure 11.** Vein-disseminated mineralization. (**a**) Pyrite grains and Cr-spinels in hydrothermally altered enstatite. (**b**) Pyrite grains and Cr-spinels in hydrothermally altered enstatite. BSE image.

Cr-Spinels from St. 242

Chromian spinel (Cr-spinels) is a common accessory in peridotitic rocks and their alteration during the serpentinization of peridotites with the formation and alteration of ferritchromite (±magnetite) rims on primary Cr-spinels grains [30]. The composition of Cr-spinels from st. 242 is not homogeneous due to significant variations in Mg, Al, Fe and Cr contents (Table 5).

**Table 5.** Variations in Cr-spinels composition from st. 242 (wt %).

| | O | Mg | Al | Cr | Fe | Si | Ti | V | Mn | Cu | Zn | S | Frequency of Cr-Spinels Occurrence in | |
| --- | --- | --- | --- | --- | --- | --- | --- | --- | --- | --- | --- | --- | --- | --- |
| | | | | | | | | | | | | | Py Layer | Cph-Py Layer |
| | | | | Primary Cr-spinels from peridotites [12,13] | | | | | | | | | | |
| Average | 35.88 | 8.08 | 14.99 | 26.35 | 14.70 | - | - | - | - | - | - | - | | |
| | | | | Primary Cr-spinels from SMS: Fe = 10%–17%; Mg + Al > 20%; Cr/Fe ≥ 2; *n* = 59 | | | | | | | | | | |
| Average | 32.77 | 8.83 | 15.97 | 28.72 | 13.49 | - | - | - | - | - | - | | | |
| Min | 31.10 | 6.37 | 13.49 | 23.48 | 11.09 | - | - | - | - | - | - | - | 80% | 36% |
| Max | 34.92 | 9.80 | 18.32 | 31.90 | 19.85 | - | - | - | - | - | - | - | | |
| | | | | Slightly hydrothermally altered Cr-spinels from SMS: Fe = 17%–26%; Mg + Al = 10%–20%; Cr/Fe = 1–2; *n* = 26 | | | | | | | | | | |
| Average | 30.55 | 5.10 | 9.69 | 32.66 | 21.31 | - | - | - | - | - | - | | | |
| Min | 28.29 | 3.06 | 6.05 | 27.16 | 17.54 | - | - | - | - | - | - | - | 7% | 24% |
| Max | 33.74 | 6.81 | 14.05 | 39.74 | 27.00 | - | - | - | - | - | - | - | | |
| | | | | Strong hydrothermally altered Cr-spinels from SMS: Fe > 26%; Mg + Al < 10%; Cr/Fe < 1; *n* = 43 | | | | | | | | | | |
| Average | 27.35 | 1.77 | 3.36 | 28.97 | 33.50 | 1.80 | 0.50 | 0.45 | 1.65 | 2.19 | 1.53 | 1.20 | | |
| Min | 23.35 | 0.31 | 0.50 | 20.07 | 21.95 | 0.13 | 0.21 | 0.22 | 0.69 | 0.14 | 0.64 | 0.23 | 13% | 40% |
| Max | 31.85 | 4.64 | 8.15 | 36.81 | 48.12 | 9.11 | 1.19 | 0.85 | 3.77 | 7.94 | 2.85 | 5.19 | | |
| | Frequency of trace elements occurrence, vol.% | | | | | 77 | 58 | 42 | 58 | 42 | 44 | 60 | | |

Notes: - not detected.

The grains of Cr-spinels are mainly characterized by a zonal structure that is represented by a core and an alteration of the outer rim (Figure 6b). There are some grains characterized by a porous, altered texture without the zonal structure (Figure 9). Correlation analysis indicates that Mg and Al have a strong correlation (coefficient corr. + 0.97), but Fe has a negative relationship with them (−0.95 and −0.94, respectively). The relationship between elements are clearly shown in the graph Mg + Al versus Fe (Table 5, Figure 12). Cr-spinels can be separated by variations of Mg, Al and Fe contents into three groups.

Cr-spinels of the group I are characterized by a high Mg content (avg. = 8.79 wt %) and Al (avg. = 15.64 wt %), low Fe (avg. = 13.63 wt %) and correspond completely to the primary peridotite Cr-spinels [12,13,30]. Trace elements were not detected. This type comprises the only core of the grains examined that predominate in the pyrite layer.

Cr-spinels from group II are characterized by Mg decreases (avg. = 4.99 wt %) and Al (avg. = 9.55 wt %) under increasing Fe (avg. = 21.85 wt %). Trace elements were not detected. This type was mainly found in slightly altered outer rims. Cr-spinels of group II are mainly related to the chalcopyrite-pyrite layer.

Cr-spinels of group III are characterized by the highest Fe content (avg. = 33.78 wt %) and the lowest contents of Mg (avg. = 1.22 wt %) and Al (avg. = 3.07 wt %). In some samples, the concentrations of Mg and Al are 0.01 wt %. Half of analyzed Cr-spinels are enriched in Si (avg. = 1.83 wt %), Ti (0.5 wt %), Mn (1.94 wt %), V (0.44 wt %), Cu (2.19 wt %), Zn (1.5 wt %) and Cl (0.89 wt %). This type of Cr-spinel is composed of porous highly altered grains and predominates in the chalcopyrite-pyrite layer.

All Cr-spinels examined could be attributed to the primary unaltered and secondary hydrothermally altered types. Secondary hydrothermally altered Cr-spinels are characterized by a distinct degree of alteration, which is reflected in their composition. Hydrothermal alteration initiates a progressive Mg and Al loss at increasing Fe and is formed owing to

very extreme conditions (such as higher fluid/rock rations, more oxidizing fluids and/or more prolonged fluid-rock interaction) [30].

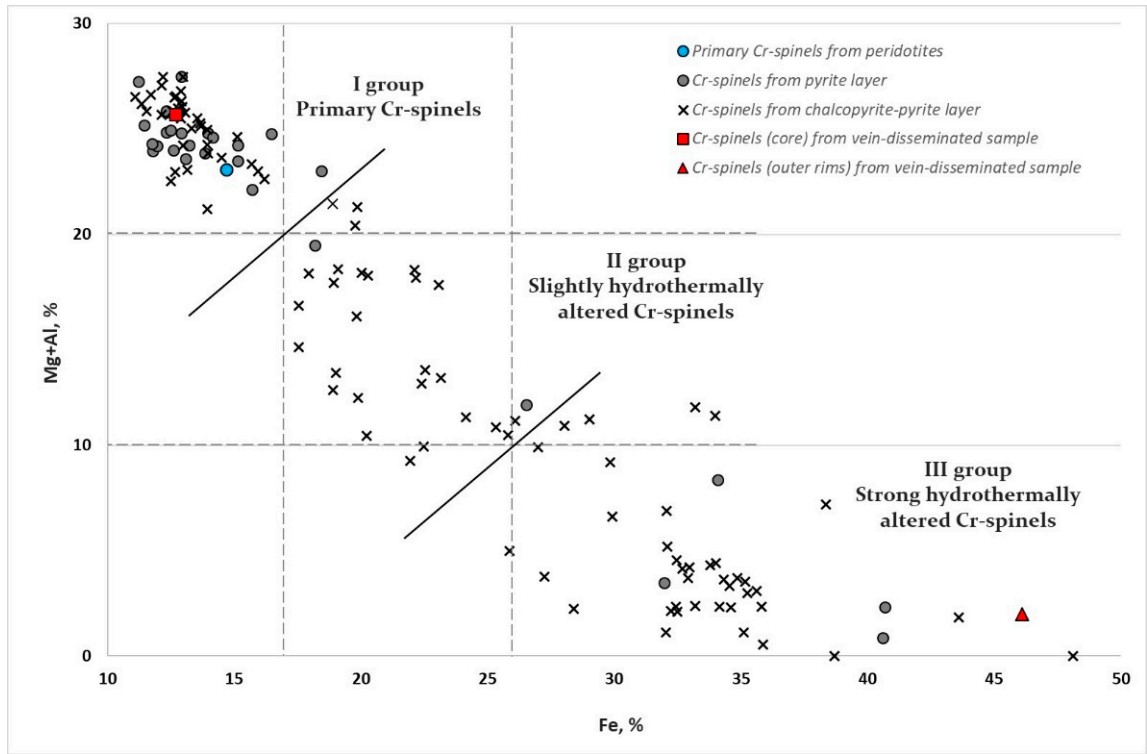

**Figure 12.** Variations of Cr-spinels composition in axes Mg + Al and Fe. Group I (primary Cr-spinels) Fe = 10–17 wt %, Mg + Al > 20 wt %, Cr/Fe > 2; group II (slight hydrothermally altered Cr-spinels) Mg + Al = 10–20 wt %, Fe = 17–26 wt %, Cr/Fe = 1–2; group III (strong hydrothermally altered Cr-spinels) Mg + Al < 10 wt %, Fe > 26 wt %, Cr/Fe < 1. The data of primary Cr-spinels from peridotites (in blue) from [12,13].

### 4.1.2. St. 372

The recovered massive sulfides are very porous (Figure 13). The major mineral is pyrite, and the minor consists of Fe oxides and hydroxides. There are insignificant amounts of marcasite and baryte. Pyrite forms thin subparallel dendritic aggregates up to 4 cm.

These samples are characterized by abundant Fe oxides and hydroxides. Moreover, Fe oxides are primary and younger than the pyrite ones. The central part of the dendrite is composed of bluish-grey needle hematite. Hematite is replaced by goethite (Figure 14a).

Fe oxides are overgrown by subhedral crystals of pyrite (up to 0.05 mm). Reniform aggregates occur occasionally (Figure 14b). Their central part contains concentrically layered or radially radiant hematite and goethite. These structures are covered by cuboctahedral pyrite. There are small relic of pyrrhotite as well as marcasite grains and baryte plates in the voids between pyrite dendrites (Figure 14c). Sulfide samples contain accumulations of mineralized tube worms (Figure 14d).

### 4.1.3. St. 373

The sampled sulfides are represented by porous pyrite and marcasite (Figure 15). Pyrite and marcasite aggregates are nodulated, scalloped and are presented in approximately equal amounts. The admixture of baryte sometimes reaches 15%. Despite a simple mineral composition, the samples are quite heterogeneous. There is abundant biogenic material. Some samples contain small fragments of basalt and sediment. Each void has a distinct rim composition.

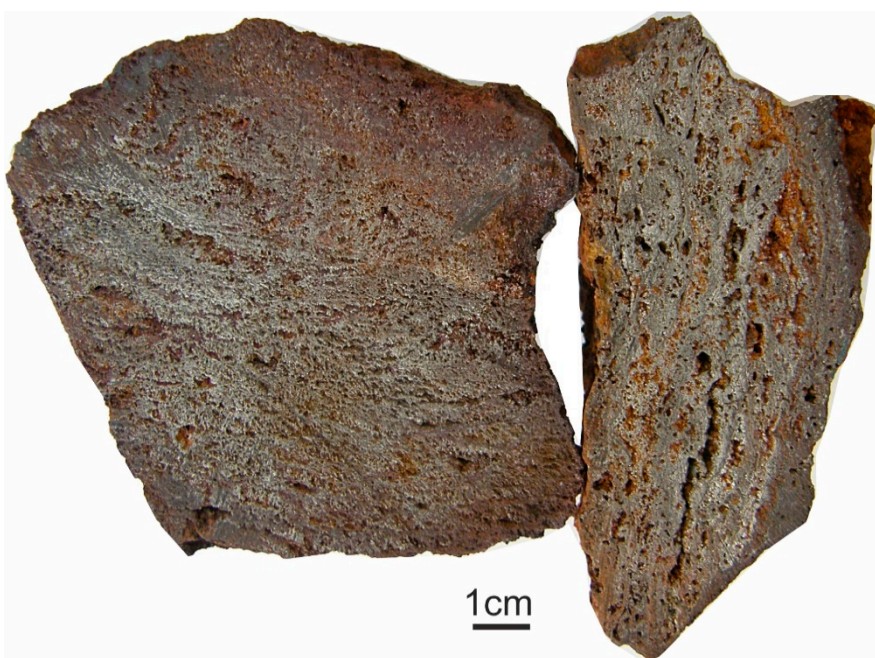

**Figure 13.** Massive porous sulfides from st. 372.

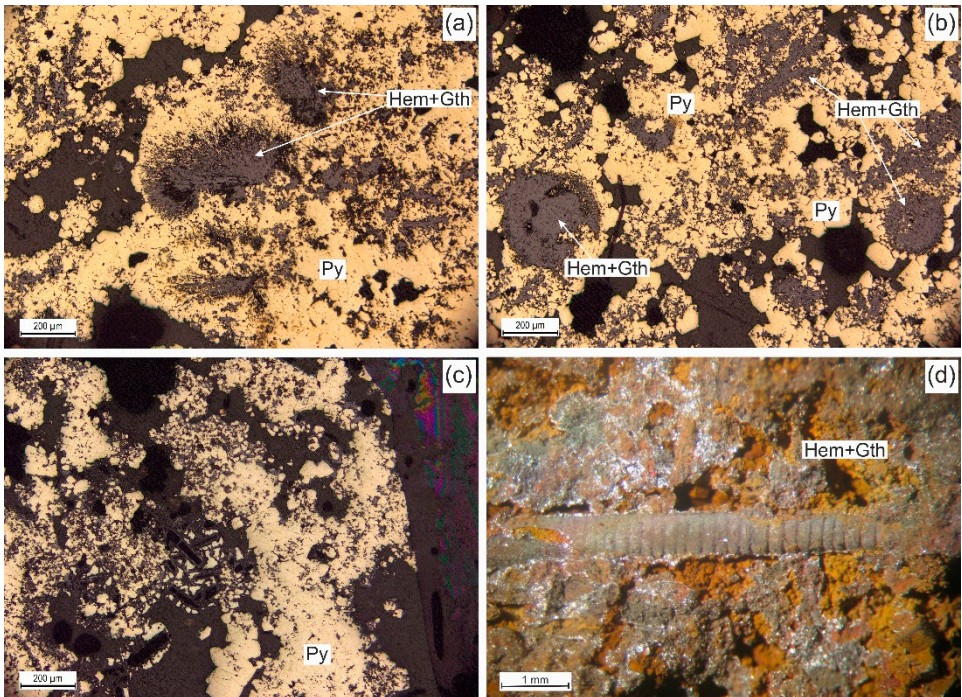

**Figure 14.** Major and minor minerals of a sulfide sample from st. 372. (**a**) Hematite aggregates in the central part of pyrite dendrites (sample 372-1/1). (**b**) Concentrically layered hematite aggregates in the central part of kidney-shaped pyrite grains (sample 372-1/1). (**c**) Relic pyrrhotite overgrown by marcasite (sample 372-1). (**d**) Mineralized tube worm in sulfides of st. 372.

Some voids contain pyrrhotite replaced by Fe oxides and rare isocubanite. Pyrrhotite seems to have been precipitated first (Figure 16a).

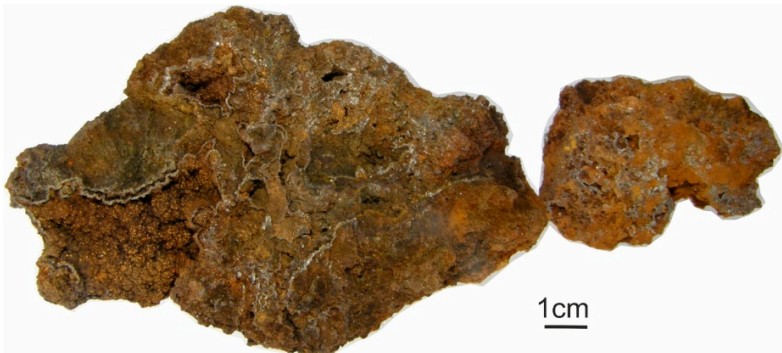

**Figure 15.** A general view of sulfide samples from st. 373. Porous massive sulfides oxidized from the surface.

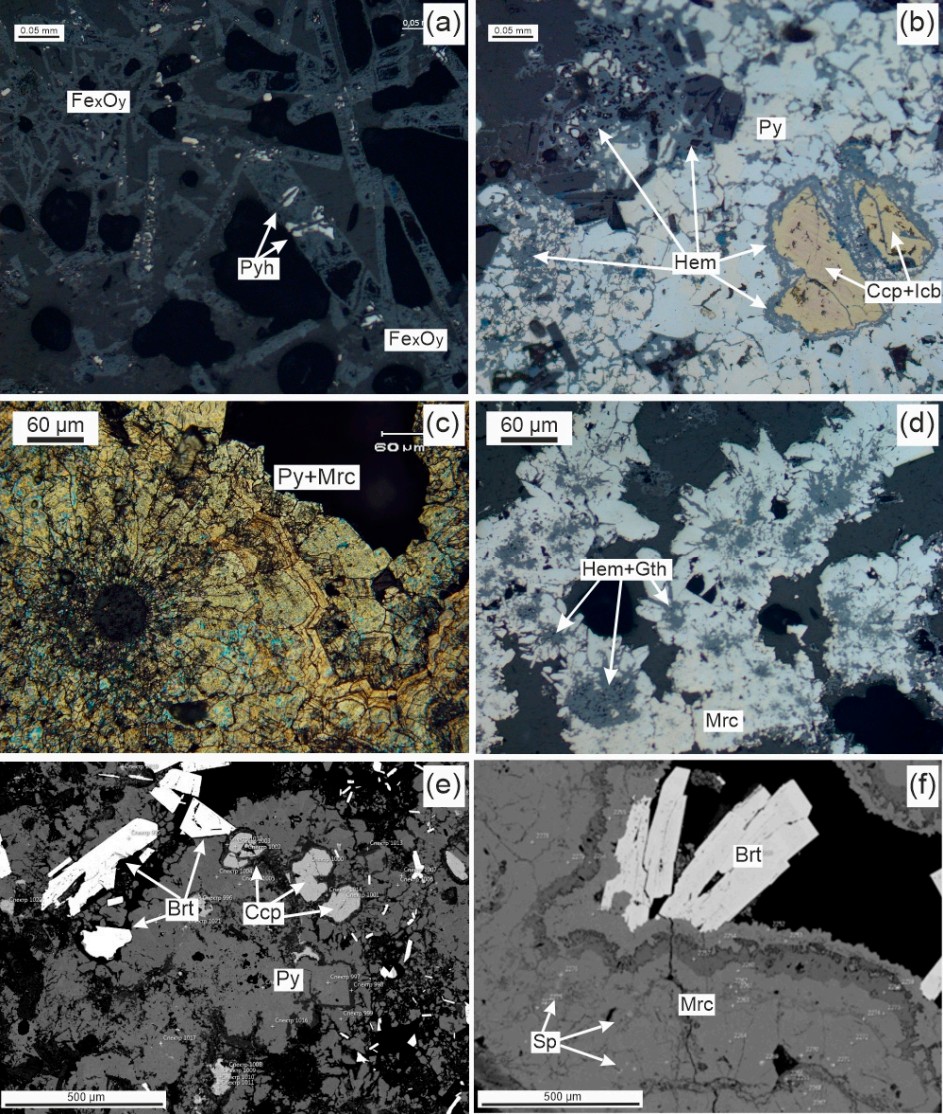

**Figure 16.** Textures and structures of sulfide samples from st. 373. (**a**) Relic pyrrhotite tabulars replaced by goethite in cavity (sample 373-4). BSE image. (**b**) Chalcopyrite with isocubanite bordered by hematite (sample 373-4). BSE image. (**c**) Radially layered aggregates of marcasite (sample 373-3/1a). (**d**) Intergrowths of lamellar hematite in the center of marcasite dendrites (sample 373-4). BSE image. (**e**) Baryte intergrowth with sulfide minerals. BSE image. (**f**) Dissemination of sphalerite in marcasite. BSE image.

Isocubanite with a chalcopyrite exsolution covered by hematite rims has been observed (Figure 16b). Pyrite varies from fine-grained (massive aggregates) to idiomorphic cubic (voids). Pyrite is often intergrown and layered with marcasite, forming radial-layered and scalloped-banded aggregates (Figure 16c). Long marcasite dendrites also occur. The central portion is comprised of hematite-goethite (Figure 16d). Granular marcasite forms intergrowths with baryte (Figure 16e). Sphalerite grains are sometimes found along the zones in pyrite crystals and abundant grains disseminated in marcasite (Figure 16f).

The samples from station 373 contain an abundance of Fe oxides, and they exhibit unusual relationships with sulfides at this station as well. The samples contain an outer oxidized zone, where sulfides are sometimes completely replaced by goethite and lepidocrocite. However, oxides are also found in the center of buds and dendrites, where sulfides grow on them, and they are on occasion distinctly interbedded with sulfides (Figure 17a). As shown above, a hematite rim separates chalcopyrite-isocubanite intergrowths from pyrite-marcasite aggregates. Sometimes hematite is contained as an interlayer in rhythmically layered marcasite, bordering the cavity (Figure 17b). There are also goethite dendrites in cavities.

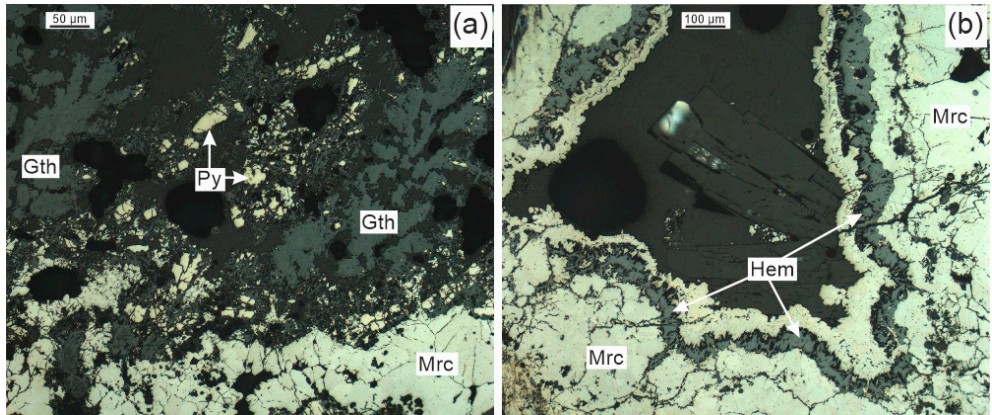

**Figure 17.** Fe-Oxides in sulfide samples from st. 373. BSE images. (**a**) Hematite-goethite dendrites in the cavity (sample 373-3/1). (**b**) Interlayer of hematite in rhythmically layered marcasite. In the center of the cavity—baryte crystals (sample 373-3/1).

The content of baryte is up to 15–20 wt% in some samples. Its idiomorphic, transparent plates reach several mm and usually form drusen. This is isolated in cavities, but also grows together with marcasite and sometimes with pyrite. The chemical composition of minerals is shown in Table 6.

**Table 6.** Average chemical composition of sulfides from st. 373.

| Mineral (Number of Analyses, *n*) | Element (wt %) | | | | | | | |
|---|---|---|---|---|---|---|---|---|
| | Fe | Cu | Zn | S | O | Ba | Sr | Total |
| Pyrite * (108) | 47.33 | - | - | 52.61 | - | - | - | 99.94 |
| Chalcopyrite (32) | 31.97 | 33.09 | - | 34.80 | - | - | - | 99.86 |
| Secondary Cu sulfides (24) | 6.73 | 59.87 | - | 31.23 | 4.59 | - | - | 100.42 |
| Fe-sphalerite (12) | 14.42 | - | 49.15 | 35.72 | 2.34 | - | - | 101.63 |
| Baryte (11) | 1.45 | - | - | 14.41 | 22.76 | 60.86 | 1.55 | 101.03 |

Notes: - not detected. * Pyrite does not contain Ni and Co.

### 4.2. Bulk Geochemistry

Similar to the previous mineralogical section, the bulk geochemistry of sulfides from the Semenov-5 hydrothermal field presented here separately for the samples recovered from the seafloor (st. 372 and 373) and sub-seafloor (st. 242) part of the deposits (Table 7). To compare our data with other NEq MAR deposits, we also present here element con-

centrations for basalt- and ultramafic-hosted SMS from the VNIIOkeangeologia database (Table 8).

**Table 7.** The chemical composition of sulfide samples from Semenov-5 hydrothermal field.

| Station | St. 242 | | | | | | | | | | St. 372 | | | St. 373 | | | | | Avg |
|---|---|---|---|---|---|---|---|---|---|---|---|---|---|---|---|---|---|---|---|
| Sample | 1 | 1/1 | 1/1b | 1/2 | 1/3 | 1/3a | 2 | 2/1 | 2/2 | 2/3 | 1 | 1 | 2 | 3 | 3/1 | 4 | 5 | 6 | |
| Fe, wt % | 44.5 | 39.5 | - | 36.6 | 35.8 | - | 25.4 | 25.0 | 25.4 | 25.3 | 44.7 | 42.4 | - | 43.1 | 41.0 | - | - | - | 35.7 |
| S | 34.2 | 45.1 | 44.0 | 34.2 | 29.1 | 42.8 | 23.2 | 28.0 | 29.7 | 31.9 | 39.9 | 36.5 | 46.2 | 38.6 | 35.7 | 39.2 | 45.8 | 44.8 | 37.2 |
| Cu | 1.39 | 7.62 | 12.9 | 13.1 | 16.6 | 13.9 | 12.0 | 11.8 | 11.9 | 12.9 | 0.11 | 1.38 | 0.79 | 1.27 | 0.71 | 1.27 | 1.03 | 0.99 | 6.76 |
| Zn | 0.02 | 0.06 | 0.10 | 0.09 | 0.13 | 0.10 | 0.06 | 0.07 | 0.08 | 0.06 | 0.05 | 0.11 | 0.09 | 0.19 | 0.07 | 0.15 | 0.03 | 0.07 | 0.09 |
| Mg | 0.19 | 0.13 | - | 0.04 | 0.05 | - | 0.22 | 0.05 | 0.07 | 0.01 | 0.04 | 0.05 | - | 0.06 | 0.04 | - | - | - | 0.08 |
| Al | 0.01 | - | - | 0.01 | 0.03 | - | 0.01 | 0.01 | 0.01 | 0.01 | 0.02 | 0.06 | - | 0.06 | 0.03 | - | - | - | 0.02 |
| Si | 1.01 | 1.48 | - | 2.11 | 2.08 | - | 14.6 | 16.2 | 14.3 | 12.6 | 0.41 | 0.54 | - | 0.54 | 0.34 | - | - | - | 5.52 |
| Ca | 0.08 | 0.08 | - | 0.01 | 0.05 | - | 0.03 | 0.01 | 0.01 | 0.01 | 0.01 | 0.01 | - | 0.01 | 0.04 | - | - | - | 0.03 |
| Co, ppm | 1150 | 1220 | 1020 | 1230 | 1090 | 1210 | 859 | 1600 | 1540 | 1170 | 191 | 9.3 | 234 | 242 | 220 | 509 | 132 | 218 | 769 |
| Ni | 1510 | 1610 | 1330 | 1600 | 1450 | 1540 | 958 | 1390 | 1110 | 1000 | 6.4 | 4.0 | 36 | 44 | 22 | 45 | 22 | 26 | 761 |
| Bi | 0.8 | 1.4 | 1.4 | 1.1 | 1.0 | 0.7 | 0.3 | 0.5 | 0.6 | 0.4 | 1.9 | <0.1 | 0.4 | 3.3 | 2 | 5.6 | 2.9 | 5.7 | 1.8 |
| Se | 482 | 248 | 382 | 370 | 270 | 483 | 102 | 415 | 358 | 327 | 451 | 6.8 | 34 | 50 | 38 | 95 | 26 | 47 | 232 |
| Te | 12 | 9.7 | 8.4 | 7.5 | 4.5 | 11 | 2.7 | 6.8 | 6.3 | 6.2 | 34 | <0.2 | 0.4 | 1.9 | 3.8 | 2.7 | 1.5 | 0.9 | 7.1 |
| In | 11 | 4.9 | 8.2 | 10 | 11 | 10 | 2.9 | 8.4 | 8.4 | 9.1 | 0.2 | 0.2 | 0.7 | 1.7 | 38 | 1.5 | 1.4 | 1.2 | 7.2 |
| Au | 0.11 | 0.14 | 0.09 | 0.12 | 0.01 | 0.13 | 0.05 | 0.10 | 0.12 | 0.07 | 1.70 | 0.17 | 0.12 | 0.38 | 0.40 | 0.49 | 0.24 | 0.13 | 0.25 |
| Ag | 23 | 36 | 21 | 33 | 23 | 28 | 7.6 | 17 | 24 | 24 | 24 | 3.2 | 6.5 | 19.3 | 12 | 13 | 18 | 14 | 19 |
| Cd | 3.3 | 2.9 | 3.1 | 2.9 | 4.4 | 3.8 | 1.2 | 2.8 | 2.8 | 3.0 | 2.9 | 0.1 | 1.1 | 0.8 | 11 | 0.6 | 0.6 | 0.7 | 2.7 |
| Ge | 0.7 | 0.5 | 0.6 | 0.6 | 0.5 | 0.7 | 0.3 | 0.6 | 0.6 | 0.6 | 11 | 0.8 | 1.2 | 2.9 | 52 | 3.1 | 2.3 | 1.9 | 4.5 |
| Ga | 1.0 | 1.1 | 0.3 | 1.3 | 1.0 | 1.3 | 2.9 | 0.7 | 0.8 | 0.7 | 4.2 | 4.2 | 1.3 | 5.1 | 10 | 2.5 | 9.4 | 2.1 | 2.8 |
| Pb | 32 | 47 | 29 | 34 | 23 | 44 | 23 | 9.6 | 20 | 25 | 116 | 184 | 508 | 224 | 93 | 301 | 217 | 330 | 125 |
| Sb | 2.0 | 3.0 | 1.8 | 3.0 | 1.2 | 3.0 | 1.3 | 0.5 | 1.2 | 1.5 | 4.9 | 2.6 | 6.9 | 9.6 | 1.5 | 8.9 | 5.9 | 7.9 | 3.7 |
| As | 27 | 35 | 25 | 38 | 21 | 34 | 23 | 6.9 | 9.6 | 17 | 48 | 117 | 103 | 225 | 4.4 | 303 | 112 | 111 | 70 |
| Sn | 18 | 19 | 14 | 18 | 12 | 19 | 8.3 | 3.5 | 6.6 | 15 | 13 | 2.3 | 2.3 | 9.1 | 10 | 6.2 | 12 | 5.8 | 11 |
| Mo | 10 | 11 | 9.1 | 8.8 | 5.3 | 14 | 2.3 | 1.1 | 1.8 | 2.5 | 7.3 | 80 | 176 | 120 | 39 | 165 | 117 | 199 | 54 |
| V | 14 | 12 | 13 | 12 | 7.7 | 15 | 9.3 | 13 | 14 | 6.6 | 2.6 | 2.7 | 10 | 32 | - | 18 | 15 | 21 | 13 |
| Cr | 1510 | 980 | 1120 | 894 | 780 | 1220 | 538 | 1020 | 1090 | 584 | 4.6 | 7.9 | 23 | 11 | - | 4.8 | 12 | 19 | 578 |
| U | <0.1 | 0.1 | <0.1 | <0.1 | <0.1 | <0.1 | 0.13 | <0.1 | <0.1 | <0.1 | 1.1 | 3.1 | 14 | 25 | 37 | 13 | 11 | 15 | 12 |
| Ba | 28 | 9.1 | 11 | 9.9 | 61 | 15 | 236 | <3 | 41 | 16 | 1220 | 11900 | 7890 | 3560 | 20 | 35100 | 16100 | 6200 | 4848 |

**Table 8.** Average content of elements in sulfide samples from Semenov-5 and another SMS at the MAR.

| | Host Rocks | | | | | | | | | | | | | SMS MAR * |
|---|---|---|---|---|---|---|---|---|---|---|---|---|---|---|
| | Mafic | | | | Mafic and Ultramafic | | | | | | Ultramafic | | | |
| Elements | Jubil-eynoye | Zenith-Victoria | Kra-snov | Seme-nov-4 | Seme-nov-5 | | | Seme-nov-2 | Seme-nov-1 | Logat-chev-1 | Irinov-skoye | Asha-dze-1 | Asha-dze-2 | |
| | | | | | St. 242 | St. 372 | St. 373 | | | | | | | |
| Fe, % | 31.5 | 39.1 | 39.8 | 41.2 | 32.1 | 44.7 | 42.7 | 13.9 | 29.3 | 19.4 | 18.0 | 28.3 | 32.0 | 32.4 |
| S | 35.7 | 44.5 | 45.3 | 47.7 | 31.3 | 39.9 | 36.9 | 14.8 | 40.0 | 23.3 | 26.1 | 28.9 | 32.3 | 36.4 |
| Cu | 4.44 | 3.02 | 1.84 | 0.93 | **11.0** | 0.11 | 1.12 | 30.1 | 4.6 | 34.1 | 21.8 | 10.2 | 18.2 | 9.56 |
| Zn | 0.45 | 1.41 | 0.63 | 0.09 | 0.07 | 0.05 | 0.12 | 3.46 | 0.16 | 2.33 | 5.16 | 17.94 | 0.86 | 4.39 |
| Si | 10.4 | 2.44 | 1.62 | 0.68 | **8.11** | 0.41 | 0.47 | 5.51 | 0.47 | 2.04 | 11.18 | 0.71 | 0.51 | 3.2 |
| Ca | 0.26 | 0.03 | 0.09 | 0.15 | 0.03 | 0.01 | 0.02 | 3.38 | 0.02 | 1.12 | 0.27 | 0.49 | 0.11 | 0.42 |
| Mg | 0.34 | 0.01 | 0.04 | 0.06 | 0.09 | 0.04 | 0.05 | 0.32 | 0.03 | 0.37 | 0.17 | 0.22 | 0.81 | 0.14 |
| Al | 0.11 | 0.12 | 0.13 | 0.14 | 0.03 | 0.08 | 0.05 | 0.22 | 0.08 | 0.21 | 0.29 | 0.16 | 0.19 | 0.16 |
| Ti, ppm | 298 | 269 | 591 | 372 | **465** | 240 | 240 | 966 | 270 | 493 | 432 | 460 | 559 | 450 |
| Co | 526 | 342 | 539 | 113 | **1267** | 191 | 223 | 92 | 48 | 467 | 150 | 2039 | 1223 | 619 |
| Ni | 6.1 | 5.1 | 6.1 | <10 | **1446** | 6.4 | 29.4 | 11 | <10 | 76 | 11 | 234 | 24 | 55 |
| Bi | 1.6 | 1.3 | 1.5 | 0.75 | 0.85 | 1.87 | **2.99** | 18 | 0.49 | 7.4 | 6.4 | 2.7 | 2.6 | 4.5 |
| Se | 355 | 11.3 | 71.4 | 101 | 383 | **451** | 43 | 409 | 130 | 276 | 365 | 229 | 89.5 | 280 |
| Pb | 61 | 161 | 47 | 81 | 28.3 | 116 | **294** | 246 | 41 | 145 | 201 | 274 | 14 | 157 |
| Ga | 15 | 12 | 7.7 | 3.2 | 0.97 | 4.2 | 4.1 | 36 | 9.2 | 7.5 | 5.1 | 7.4 | 7.0 | 19 |
| Ge | 2.1 | 14 | 3.3 | 3.6 | 0.62 | **11.5** | 2.0 | 345 | 24 | 9.1 | 35 | 9,3 | 1.7 | 36 |
| Sn | 5.8 | 2.5 | 24 | 24 | 13.4 | 12.8 | 6.3 | 108 | 26 | 132 | 44 | 368 | 40 | 65 |
| Cd | 8.3 | 22 | 26 | 0.9 | 3.1 | 2.9 | 0.67 | 113 | 3 | 77 | 129 | 287 | 13 | 110 |
| Sb | 6.1 | 34 | 9.1 | 3.8 | 1.9 | 4.9 | **6.9** | 17 | 10 | 152 | 90 | 51 | 10.9 | 33 |
| As | 19 | 188 | 54 | 140 | 24 | 47 | **161** | 106 | 6.6 | 386 | 356 | 84 | 215 | 318 |
| Mn | 388 | 54 | 29 | 15 | **44.2** | 8.0 | 18 | 86 | 215 | 245 | 164 | 473 | 108 | 159 |
| Ba | 1451 | 450 | 8623 | 6200 | 21.5 | 1220 | **13,458** | 1144 | 10,045 | 1900 | 218 | 690 | 300 | 2900 |
| Ag | 15 | 28 | 25 | 11 | 26 | 24 | 12.4 | 185 | 20 | 46 | 86 | 84 | 7.5 | 49 |
| Au | 0.5 | 0.3 | 0.8 | 0.4 | 0.1 | **1.7** | 0.2 | 25 | 5.2 | 14 | 5.2 | 3.3 | 11 | 3.2 |
| Cr | 88 | 30 | 38 | 47 | **984** | 4.6 | 12.8 | 36 | 2.5 | 38 | 129 | 24 | 14 | 38 |
| Mo | 147 | 126 | 139 | 88 | 6.7 | 7.3 | **143** | 167 | 145 | 163 | 25 | 43 | 80 | 97 |
| Te | 11 | 1.2 | 4.3 | 1.0 | 8.5 | **35** | 1.2 | 18.3 | 3.6 | 15 | 16 | 7.7 | 14 | 9.9 |
| In | 3.8 | 1.4 | 2.5 | 1.5 | **8.8** | 0.2 | 1.1 | 0.8 | 0.6 | 6.8 | 3.2 | 9.3 | 3.7 | 4.4 |
| V | 49 | 50 | 15 | 70 | 12 | 2.6 | 16 | 55 | 341 | 187 | 69 | 57 | 258 | 160 |
| n | 35 | 63 | 161 | 99 | 9 | 1 | 6 | 35 | 11 | 90 | 10 | 120 | 53 | 1052 |

Note: Bold—higher than average on Mid-Atlantic Ridge (MAR) sulfides; n—Number of samples; * Average value of elements for MAR SMS from VNIIOkeangeologia database.

### 4.2.1. St. 242

Sulfide samples from st. 242 are characterized by high Co (1267 ppm), Ni (1446 ppm) and Cr (984 ppm) as well as Se (383 ppm) and In (8.8 ppm) average contents compared with sulfides of the MAR. There are low Au concentrations (0.11 ppm). Massive sulfides of the pyrite and chalcopyrite-pyrite zones differ in Cu (7.4% and 13.1%) and especially in Si (1.5% and 12%) content, respectively.

### 4.2.2. St. 372

St. 372 is represented by only one sample with a pyrite composition. There is low Cu and Zn content (Cu—0.10% and 0.05%), but Zn content, as well as As, Sb, Mo, Pb, Ba and U are higher than in the pyrite samples of st. 242. Co (191 ppm), Ni (6.4 ppm) and Cr (4.6 ppm) values are relatively low. However, the Se concentrations (451 ppm), Te (34.7 ppm) and Au (1.7 ppm) are the highest for the Semenov-5 sulfides.

### 4.2.3. St. 373

In the marcasite-pyrite samples, Cu content (1.12 ppm) is almost the same as in thepyrite samples from st. 242. Although Zn concentration is not so high (0.12 ppm), it is at its maximum for the Semenov-5 field. Sulfide samples from st. 373 are enriched in As (161 ppm), Mo (143 ppm), Pb (294 ppm) and especially in U (13.5 ppm) and Ba (1.35%). It should be noted that Co (223 ppm), Ni (29.4 ppm) and Cr (12.8 ppm) are elevated here relative to sulfides from st. 372.

The comparison means values of element concentrations from the Semenov-5 and other SMS deposits within the NEq MAR demonstrate strong enrichment in Co, Ni and Cr, elevation in Cu and depletion in Zn, Au and Ag (Table 8).

## 5. Discussion

Modern SMS as well as ancient volcanogenic massive sulfides (VMS) deposits are generally composed of massive and disseminated sulfide zones. Massive sulfides form mound-like structures on the seafloor surface as a result of a reaction between discharged hot hydrothermal fluid and cold seawater. Disseminated sulfides fill the cracks in host rock in the stockwork zone below the seafloor. The most problematic issue is the extension of massive sulfides to the sub-seafloor, which was generally suggested for the ultramafic SMS deposits by Y. Fouquet et al. (2010) [31] and later for the TAG field by B. Murton et al. (2019) [10]. However, the parameters surrounding the presence of subsurface massive sulfides are still uncertain. Data from the Semenov-5 field provide new information regarding this issue [3,4,9].

The inner/subsurface part of the Semenov-5 field was exposed as a result of a mass wasting process and a landslide on the northern slope of the dome-shaped structure. These conditions persisted for an entire Semenov SMS deposit (Figure 18) [14,20]. Dredging on the outcropped sub-seafloor zone led to recovery of massive sulfide mineralization. The mineralogy and geochemistry of the sulfides sampled give evidence that they have been formed by replacement of basement (ultramafic) rocks and the metasomatic nature of the sub-bottom part of deposit.

The mineralogical indicator of the replacement process could be relict Cr-spinel grains in different degrees of alteration identified in sulfide samples from st. 242. Cr-spinel is a typical mineral for ultrabasic rocks and the presence of Cr-spinels in massive sulfides indicates that ore mineralization was formed as a result of host rock replacement. Similar observations interpreted as replacement/metasomatism of hosted peridotites have been revealed for Cr-spinels in Early Devonian VMS deposits (e.g., Ishkinskoye, Ivanovskoe) from the Urals [2]. It was shown that Cr-spinels in onland VMS deposits are sensitive to the hydrothermal alteration process and their composition depends on partial pressure of oxygen and temperature [32]. The investigation of Cr-spinels composition of MAR show varieties from a species with a high content of MgO and $Al_2O_3$ at low FeO to Cr-spinels with high FeO at low MgO and $Al_2O_3$, marking the first stage of oxidation. The content of

$Cr_2O_3$ is fairly constant but could drop, marking the second stage of oxidation [30]. The increasing Fe content is indicative of the process of alteration of VMS deposits along with MAR chromites [30,33]. Fe-rich alteration rims are formed because of changing conditions (such as higher fluid/rock rations, more oxidizing fluids, and/or more prolonged fluid-rock interaction). In the process of interaction between hydrothermal fluid and relict Cr-spinels from ultramafic rock, Mg and Al are removed with consequent enrichment of Fe at almost constant Cr [30].

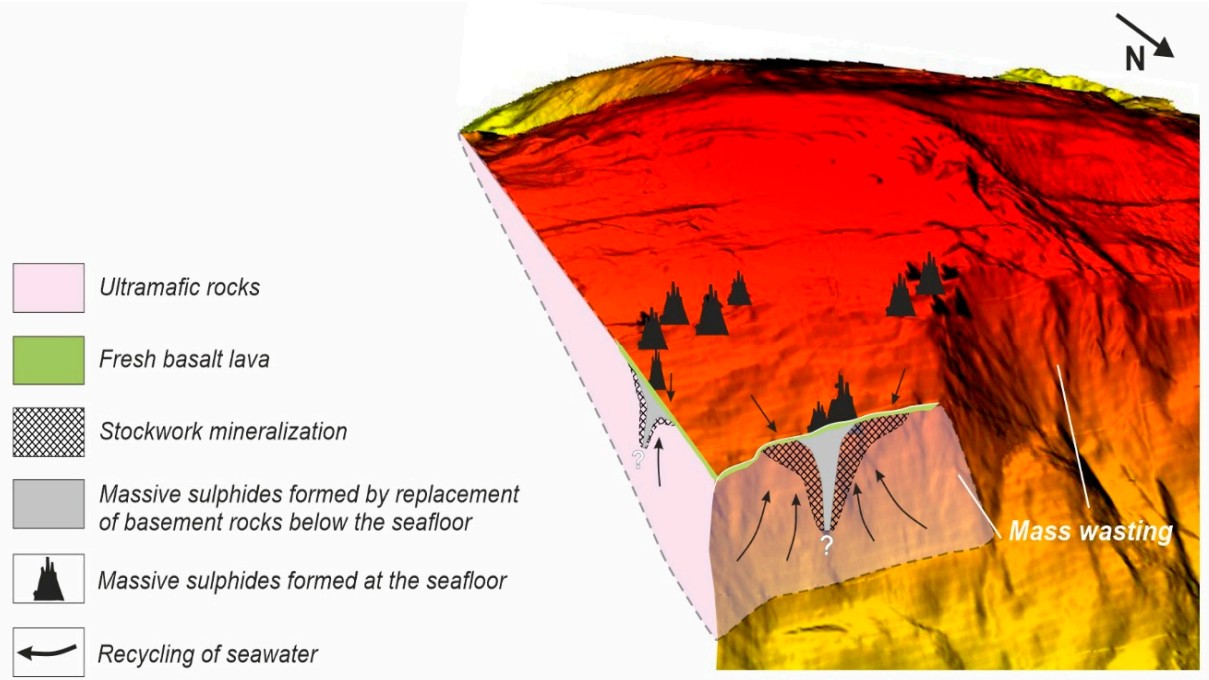

**Figure 18.** The model of inner structure of the Semenov-5 hydrothermal field. The bathymetry data is from Escartin et al. [20].

The results of our study corresponded to data described above for onland VMS deposits and MAR chromites. We detected variations of composition within the grain of Cr-spinels: the central zone is represented by primary Cr-spinels with high MgO, $Al_2O_3$ and low FeO, whereas the outer rim zone is composed of secondary Cr-spinels with high FeO and low MgO and $Al_2O_3$ content (Table 5, Figure 12). In the process of interaction between the hydrothermal fluid and relict Cr-spinels from ultramafic rock, Mg and Al are removed with consequent enrichment of Fe at almost constant Cr, which demonstrates the first stage of oxidation [30]. The presence and composition of Cr-spinels in sulfides confirm metasomatism as a part of the ore-forming process at the Semenov-5 hydrothermal field.

In terms of geochemistry, the chemical composition of subseafloor massive sulfides also gives evidence of metasomatic processes. Elevated Cu, Co and Ni are typical for ultramafic-hosted SMS and are related to the recycling process that leaches these metals from the host rocks. However, Cr concentration in the studied samples is significantly higher than the average Cr content in ultramafic hosted SMS. In comparison with the mean value for the ultramafic hosted SMS (37 ppm), the content of Cr in sulfides of st. 242 (984 ppm) is 26 times higher. Such extremely high Cr content in seafloor massive sulfides is recorded for the first time and could be related to the presence of relict Cr-spinels.

All the above mineralogical and geochemical data in evidence on the sub-seafloor massive sulfide formation in the area of st. 242 comes as a result of the replacement processes of ultramafic rocks. Morphology of the replacement zone (ore channel) could be described as "trumpet-shaped", i.e., narrow and tubular in the deep part of the ore channel and widening in the upper portion (see below). The upper widened part of replacement zone was suggested by Y.Fouquet et al. [31] and Murton et al. [10] as the

subsurface replacement zone. We propose that this upper part has a tube-shape ore channel continuation in the deep.

In addition to sub-seafloor sulfide mineralization, massive sulfides were sampled at the surface of basalt lava flows which cover utramafics close to the top of a OCC structure near st. 372 and 373. The samples recovered are composed of pyrite, Co-rich pyrite, marcasite, baryte and abundant oxyhydroxides of Fe. There are local zones enriched in isocubanite and relict pyrrhotite. The chemical composition of seafloor and sub-seafloor sulfides differs considerably (Table 7). The first ones are relatively enriched in Fe, Se and Te as well as in Ba, Pb, Mo, As and U and depleted by Cu, Co and Ni.

Differences in composition are, in fact, related to the redox parameters and temperature of the fluid. The fluid alters and replaces the rocks below the surface and, after discharge, forms massive sulfides on the seafloor. It is possible that the same fluid, forming high-copper sulfides under the bottom, discharged mainly iron on the surface where low temperature minerals formed iron sulfides (marcasite) that were enriched in trace elements from recharged seawater.

Basalts and ultramafics are usually considered as the source of metals for massive sulfides. High Ni and Co content point to an ultramafic source [31]. Elevated Se and Te more often occur in ultramafic-hosted SMS and the seafloor condition is proposed as being more favorable for Se and Te precipitation [14,31]. Based on our data [32], the U content of 13.5 ppm is too high for basalt-hosted SMS and is more typical of ultramafic hosted fields. It has also been shown for VMS deposits that seawater could be a source of Ba, Pb, Mo and As [33,34].

Therefore, we presume a greater influence of ultramafic rocks and mixing of hydrothermal fluid and seawater on the composition of massive sulfides on the surface of the seafloor than basalts. It is proposed that basalts do not effect sulfide composition because of a small volume of lava erupted on the giant massif of ultramafic rocks.

Generalizing the obtained data, the following reconstruction of the ore-forming process on Semenov-5 is proposed (Figure 19).

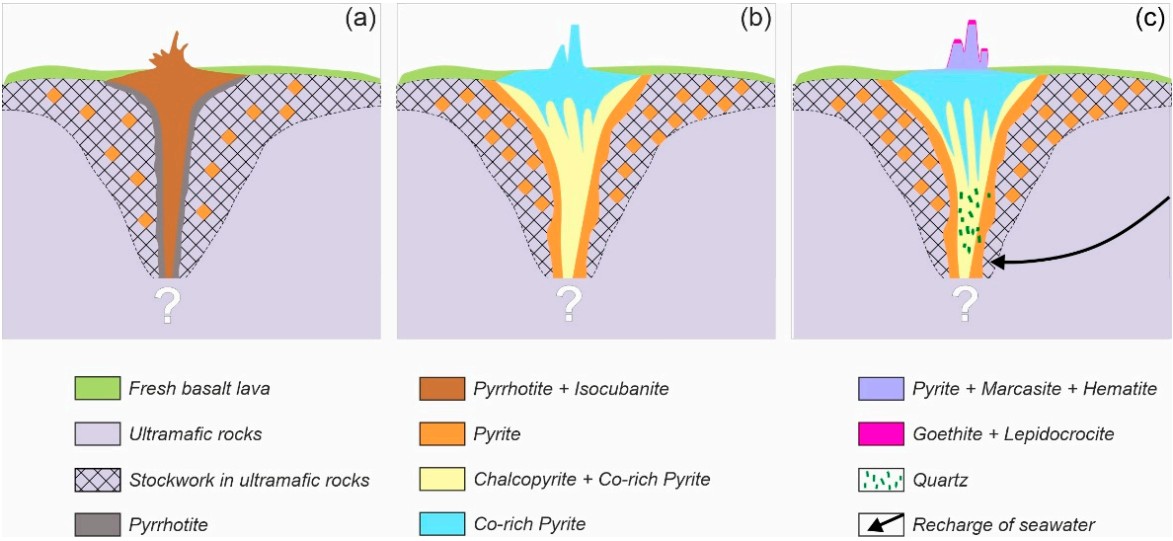

**Figure 19.** Reconstruction model for massive sulfide formation in the Semenov-5 hydrothermal field. (**a**) Sub-seafloor formation of pyrrhotite and pyrrhotite-isocubanite as a major sulfide channel by replacement of ultramafic host rock; seafloor massive sulfides are also represented by relict pyrrhotite and isocubanite. (**b**) Replacement of pyrrhotite and isocubanite by pyrite, chalcopyrite and Co-rich pyrite; seafloor deposition of Co-rich pyrite. (**c**) Sub-seafloor formation of abundant quartz with rutile and quartz with hematite as a result of sub-seafloor mixing of hydrothermal fluid with seawater; seafloor deposition of pyrite-marcasite interbedded with hematite. Goethite and lepidocrokite cover massive sulfide as a result hypergenesis.

Presumably, the initial hot hydrothermal fluid nearly leached and altered the ultramafic host rock. The high temperature sulfides (pyrrhotite and isocubanite) replaced the altered peridotites and formed a major ore channel (Figure 19a) [35]. The outer part of ore channel is composed of pyrrhotite; the central one by pyrrhotite with isocubanite. At the decrease of the fluid temperature, pyrrhotite from the outer part began to be replaced by pyrite without impurities (Figure 19b). Primary Cr-spinels without impurities are mostly preserved in this pyrite (apopyrrhotite) zone. These observations indicate that the process related to this zone formation took a short time.

At the same time, pyrrhotite and isocubanite were also precipitated on the seafloor surface (Figure 16a,b).

Further conductive cooling of the fluid leads to the sub-seafloor formation of high-temperature chalcopyrite and pyrite with elevated Co and Ni. These replaced primary isocubanite and pyrrhotite, respectively. Slightly hydrothermally altered secondary Cr-spinels have wide Fe-rich rims which point to the extended influence of hydrothermal fluid within this layer.

The massive sulfides formed from the same fluid on the seafloor surface are presented by Co-rich pyrite with elevated contents of Se, Te and Co (Figure 19b).

Furthermore, in the central zone of the major ore channel, abundant quartz with rutile and quartz with hematite formed around abundant pores and filled cracks (Figure 19c). Strongly hydrothermally altered secondary Cr-spinels are represented by the widest Fe-rich rims with an insignificant core and became very porous. These grains are enriched by Si, Ti, Mg, V and Zn. Based on the observations, we propose the presence of seawater recharge that leads to a non-equilibrium condition of the system and an increase of partial pressure of oxygen. This zone was formed by the long-term influence of hydrothermal fluids mixed with recharged seawater.

Massive sulfides on the seafloor surface formed from the mixed fluid and seawater are represented by pyrite-marcasite interbedded with hematite. The interbedding sulfides with oxides characterized an unstable variation in the regimes of sulfur and oxygen were described for several SMS deposits [30,36].

The latest secondary oxidation processes resulted in formation of lepidocrokite and goethite at the surface of sulfides (Figure 19c).

Along with sub-seafloor and seafloor massive sulfides, vein-disseminated mineralization also developed in the hosted rock. The vein-disseminated sulfides recognized as stockwork are represented by idiomorphic pyrite and sphalerite in hydrothermally altered (serpentinized) peridotites. Slightly altered Cr-spinels with a thin ferrous rim were observed in samples from the stockwork zone as well. Stockwork mineralization was generated as a result of filling cracks and pores by sulfides precipitated from circulated solutions in highly permeable serpentinites.

To sum up, three zones of a hydrothermal ore-forming system have been described: massive sulfides precipitated from hot vents on the surface of the seafloor: (1) massive sulfides formed due to the replacement of ultramafics below the seafloor, (2) disseminated sulfide mineralization-filled cracks in hosted rocks, and (3) the stockwork formed around sub-seafloor massive sulfides. The unique geological setting of the Semenov-5 hydrothermal field allowed us to make this reconstruction based on a limited number of samples. It should be emphasized that all the sampled minerals were formed as a result of fluid circulation in ultramafic rocks and linked by a common ore-forming process.

Continuation of massive sulfides below the surface of the seafloor could considerably increase the resource estimation of SMS deposits because of the presently calculated volume of ore bodies that conscribe the limits of the paleo-seafloor boundary. However, this consideration should be demonstrated by the drilling of ultramafic-hosted SMS deposits.

## 6. Conclusions

The mass wasting landslide processes at 13°30′ N OCC in the area of the Semenov-5 hydrothermal field has provided an exceptional opportunity to observe mineralization

in a complete hydrothermal ore forming system: the surface and subsurface portion of an ultramafic-hosted sulfide deposit. Sub-seafloor mineralization includes disseminated and massive sulfides, which differ from surface ones. The presence of relict Cr-spinels at a different degree of alteration as well as extremely high concentrations of Cr in massive sulfides indicate the replacement (metasomatic) nature of their formation. It is proposed, for the first time, that the morphology of the replacement zone is presented by a deep narrow tubular part, which widens closer to the seafloor surface.

Based on the mineralogy and geochemistry of the sub-seafloor and seafloor sulfides, the complex ore-forming processes of ultramafic-hosted SMS deposits were reconstructed. According to the developed model, the first stage of the ore-forming process was initiated within the ore channel on the sub-seafloor setting with alteration of hosted rocks by high temperature (400 °C) sulfides (pyrrhotite and isocubanite), later replaced by stoichiometric pyrite. The initial fluid locally reached the seafloor and the precipitated pyrrhotite and isocubanite, which are now presented as relicts. A following cooling of fluid led to the formation of chalcopyrite with Co-rich pyrite, which replaces pyrrhotite and isocubanite on the subseafloor and precipitate Co-rich pyrite with elevated Se, Te and Co on the seafloor surface. Seawater recharge into the sub-seafloor zone disturbs the equilibrium conditions of the system and results in the formation of abundant quartz with rutile and quartz with hematite precipitated from fluid mixed with seawater. On the seafloor surface, a mixed fluid and seawater precipitated pyrite-marcasite is interbedded with hematite.

Summarizing data obtained, it is concluded that

(a)   the significant differences of composition between Semenov-5's (1) sub-seafloor and (2) seafloor mineralization are most likely connected with altering the physical and chemical parameters of the hydrothermal system and a difference in the mode of formation: (1) metasomatically within hosted rocks and (2) from the discharged fluid on the seafloor surface;

(b)   sub-seafloor and seafloor massive sulfides have a common history of formation from the hydrothermal fluids which have been circulated within the ultramafic rocks and discharged on the surface;

(c)   distribution of fresh, thin basalt lava flowing within the studied area does not significantly affect seafloor massive sulfide composition.

The suggested model of inner structure allows not only for a better understanding of the sub-seafloor hydrothermal process but could be applied to more reliable resource assessments of ultramafic-hosted SMS deposits as well.

**Author Contributions:** A.F. studied the samples of massive sulfide, performed the microprobe analyses, analyzed data, interpreted results and wrote the text; T.S. and V.B. collected samples and described the samples onboard, interpreted results and wrote the text; A.S. analyzed data and wrote the text; G.C. revised the manuscript. I.P. wrote the text. All authors have read and agreed to the published version of the manuscript.

**Funding:** This research has been funded by the Russian Science Foundation research project № 22-27-00375.

**Data Availability Statement:** Not applicable.

**Acknowledgments:** The authors express their thanks to Larisa Lazareva, Irina Rozhdestvenskaya and Victor Ivanov for the provided samples and valuable advice; operators Vladimir Shylovskyh and Natalia Vlasenko for help in work on scanning electron microscope.

**Conflicts of Interest:** The authors declare no conflict of interest.

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
