# Peer review of "New Data for the Internal Structure of Ultramafic Hosted Seafloor Massive Sulfides (SMS) Deposits: Case Study of the Semenov-5 Hydrothermal Field (13°31′ N, MAR)"

_minerals, doi:10.3390/min12121593_

Round 1
Reviewer 1 Report
The concept is a good one - to look at a tectonic scarp slope that cuts down through the rock stratigraphy to expose the roots of a hydrothermal system. However, unfortunately I do not think that the authors present the data is a way that convinces you that you are looking at the sub-seafloor. The samples described are not clearly sub-seafloor and there are several features that suggest at least some of them are partially weathered seafloor chimneys. In addition:
- There are assumptions that need to be explained (e.g., temperature of py/po precipitation)
- There have been several International Ocean Discovery Program (IODP) expeditions to drill SMSs and investigate their 3rd dimension - these are not even mentioned
- The Cr Spinel story is an interesting one that could be explored further (e.g. the concept of "immobile elements" that are used for igneous rock classification
- "Methods" is unnecessarily detailed - no need to name the operators! (is some of this detail appropriate for an appendix?)

Author Response
Thank you for review of our manuscript. We are appreciate for all your comments, which helped us to improve manuscript.
"Please see the attachment." in the box

Reviewer 2 Report
The topic of the manuscript is relevant, new data are presented that add information about the genesis of SMS deposits. Manuscript is well illustrated. But there are some questions that require clarification and comments that should be responded to. Comments are shown in the peer-reviewed manuscript.
1) The abstract to the manuscript reflects its content, but exceeds the allowed length. From the instructions for authors: «The abstract should be a total of about 200 words maximum. The abstract should be a single paragraph and should follow the style of structured abstracts: Background, Methods, Results (Summarize the article's main findings), Conclusion».
1) Claims for tables:
a) In the tables of mineral compositions, there are no sums of analyzes, which is unacceptable. The meanings of some symbols should be explained: "?", "-", "x". It would be preferable to present results as representative primary analyzes rather than maximum and minimum values.
b) It is not clear how out of 198 analyzes ("n=198") both the minimum and maximum Ti and Cl values are the same. Does this mean that all 198 analyzes in pyrites from chalcopyrite-pyrite layer showed 0.94 wt.% Cl and 4.54 wt.% Ti? This is probably a mistake.
c) Analyzes of minerals containing oxygen (Enstatite, Cr-spinels) are usually presented in oxides, and not in elements
d) Pyrite, containing both chromium, titanium, and silicon in the form of trace elements at the same time in such significant quantities, is too incredible to be true. It's most likely an artifact. Analysis error. Or these results should be discussed in detail. Most likely, porous pyrite contains in its pores a mechanical admixture of other minerals (quartz, chromite, ilmenite). But at the same time, there is no oxygen in the analyzes. Unclear. If the authors have no doubts about this, then references should be made to works that describe the same pyrite.
2) The term "apopyrhotite tabular" is frequently used in the manuscript. It is possible that this is a generally accepted term, but for readers of a different field of research, an explanation should be given why the authors believe that pyrite is formed precisely from pyrrhotite, if there are no relics of this mineral. It would be more logical to assume that in some cases pyrite is formed after isocubanite, which follows from Figures 8a and 8b.
3) The mineral symbols on the figures are arbitrary. For each mineral, certain symbols are accepted by IMA: chalcopyrite - Ccp, baryte - Brt, hematite - Hem, goethite - Gth , marcasite - Mrc, rutile - Rt . You should look at https://rruff.info/ima/#
In Discussion, the authors claim that The oxidizing fluids and fluid-rock interaction have enabled Cr leaching and diffusion out of the rims, as well as the removal of Mg and Al with subsequent enrichment of chromite rims with iron, referring to the work [26 ] (Ribeiro da Costa and Barriga, 2022).
However, it was shown in (Ribeiro da Costa and Barriga, 2022)
that Cr remains in chromite when Mg and Al are removed. And it can be removed only at the final stage of replacement, when magnetite rims are formed after chromite. This is not observed in the studied ores, judging by Table 5: even in intensely altered chromite grains, the chromium concentration remains at the same level as in unaltered ones. And trace components (Mn, Ti, V, Ni and Co) seem to be relatively immobile during spinel oxidation (Ribeiro da Costa and Barriga,2022)
The question remains: where does Cr and Ti come from in pyrite?
4) If so much attention is paid to chromite and its replacement in the process of interaction with the fluid in the manuscript, then it is necessary to insert more informative diagrams on chromites, for example, those given in the cited article 26. And the Al - Cr - Fe3+ triple diagram, which would show variations on chromium.
I wish the authors success.
19.11.2022

Author Response
Dear reviewer,
Thank you for review of our manuscript. We are appreciate for all your comments, which helped us to improve manuscript.
Please see the attachment

Round 2
Reviewer 2 Report
I am satisfied with the numerous corrections made by the authors in the new version of the manuscript.